# Random Reshuffling is Not Always Better

**Christopher De Sa**
Department of Computer Science
Cornell University
cdesa@cs.cornell.edu

## Abstract

Many learning algorithms, such as stochastic gradient descent, are affected by the order in which training examples are used. It is generally believed that sampling the training examples without-replacement, also known as *random reshuffling*, causes learning algorithms to converge faster. We give a counterexample to the Operator Inequality of Noncommutative Arithmetic and Geometric Means, a longstanding conjecture that relates to the performance of random reshuffling in learning algorithms [19]. We use this to give an example of a learning task and algorithm for which with-replacement random sampling outperforms random reshuffling.

## 1 Introduction

Many machine learning algorithms work by iteratively updating a model based on one of a number of possible steps. For example, in stochastic gradient descent (SGD), each model update is performed based on a single example selected from a training dataset. The *order* in which the samples are selected—in which the update steps are performed—can have an impact on the convergence rate of the algorithm. There is a general sense in the community that the *random reshuffling* method, which selects the order by without-replacement sampling of the steps in an epoch (where an *epoch* means a single pass through the data, and different epochs may use different random orders), is better (for convergence) than ordinary with-replacement sampling for these algorithms [7, 8, 19].

There are two intuitive reasons why we might expect random reshuffling to outperform sampling with replacement. The first applies when our model updates are in some sense *noisy*: each one could perturb us away from the desired optimum, and they are only guaranteed to approach the optimum *on average*. In this case, random reshuffling ensures that the noise in some sense "cancels out" over an epoch in which all samples are used. Most previous work on random reshuffling has studied this noisy case, and this intuition has been borne out in a series of results that show random-reshuffling results in a convergence rate of $O(1/t^2)$ rather than $O(1/t)$ for convex SGD [8, 10, 16, 18, 20].

The second intuitive reason is that, because sampling without replacement avoids using the same update step repeatedly, it should tend to be "more contractive" than sampling with replacement. This intuition applies even for "noiseless" algorithms that converge at a linear rate of $O(1)^t$. In contrast to the noisy case, the belief that random reshuffling should be better in general for these algorithms that converge at a linear rate is backed up theoretically only with conjectures. The main conjecture in this space is the *Operator Inequality of Noncommutative Arithmetic and Geometric Means*, stated as Conjecture 1 of Recht and Ré [19]. That conjecture, which is motivated by algorithms such as the randomized Kaczmarz method [23] that converge at a linear rate, asserts the following.

**Conjecture 1** (Operator Inequality of Noncommutative Arithmetic and Geometric Means)**.** *Let $A_1, \dots, A_n \in \mathbb{R}^{d \times d}$ be a collection of (symmetric) positive semidefinite matrices. Then it is*

*conjectured that the following inequalities always hold:*

$$\left\| \frac{1}{n!} \sum_{\sigma \in \mathcal{P}(n)} \prod_{i=1}^{n} A_{\sigma(i)} \right\| \leq \left\| \left( \frac{1}{n} \sum_{i=1}^{n} A_i \right)^n \right\|, \tag{1}$$

$$\left\| \frac{1}{n!} \sum_{\sigma \in \mathcal{P}(n)} \left( \prod_{i=1}^{n} A_{\sigma(i)} \right)^T \left( \prod_{i=1}^{n} A_{\sigma(i)} \right) \right\| \leq \left\| \frac{1}{n^n} \sum_{f \in \{1,\ldots,n\}^n} \left( \prod_{i=1}^{n} A_{f(i)} \right)^T \left( \prod_{i=1}^{n} A_{f(i)} \right) \right\|, \tag{2}$$

*where $\mathcal{P}(n)$ denotes the set of permutations of the set $\{1, \ldots, n\}$ and $\| \cdot \|$ denotes the $\ell_2$ induced operator norm (the magnitude of the largest-magnitude eigenvalue for symmetric matrices).*

A variant of the conjecture, which moves the sums to the outside of the norms, was given by [7].

**Conjecture 2.** *Let $A_1, \ldots, A_n \in \mathbb{R}^{d \times d}$ be a collection of (symmetric) positive semidefinite matrices. Then it is conjectured that the following inequality always holds:*

$$\frac{1}{n!} \sum_{\sigma \in \mathcal{P}(n)} \left\| \prod_{i=1}^{n} A_{\sigma(i)} \right\| \leq \frac{1}{n^n} \sum_{f \in \{1,\ldots,n\}^n} \left\| \prod_{i=1}^{n} A_{f(i)} \right\|. \tag{3}$$

Conjecture 1 is a quite natural generalization of the ordinary arithmetic-mean-geometric-mean (AMGM) inequality of real numbers, which states that for non-negative numbers $x_i$,

$$\prod_{i=1}^{n} x_i \leq \left( \frac{1}{n} \sum_{i=1}^{n} x_i \right)^n.$$

In Conjecture 1, positive semidefinite matrices (matrices with non-negative eigenvalues) take the place of the non-negative scalars of the AMGM inequality, and indeed Conjecture 1 reduces to the AMGM inequality when $d = 1$. Conjecture 1 was proven by the original authors in the case of $n = 2$, and has been proven subsequently for $n = 3$ [12, 29]. It also seems to be true for random ensembles of matrices [2, 19], and random testing seems to suggest that Conjecture 1 is always true. However, recent work has shown non-constructively that Conjecture 1 is false [3, 14].[1] These non-constructive disproofs are interesting, but deliver limited insight about random reshuffling, both because they involve complicated proof techniques and because they do not translate to concrete counterexamples of matrices $A_1, A_2, \ldots, A_n$ that can be used to study learning algorithms empirically.

In this paper, we propose simple counterexamples for these conjectures—to our knowledge this is the first explicit counterexample known for any of these conjectures, and the first disproof of Conjecture 2. We explore the consequences and limitations of this counterexample throughout the paper, and end by showing concrete problems for which SGD with random reshuffling converges asymptotically slower than SGD using with-replacement sampling. Our paper is structured as follows.

- In Section 2, we construct a family of counterexamples for Conjectures 1 and 2, showing constructively that all three conjectured inequalities are false.
- In Section 3, we adapt the counterexample to give concrete ML algorithms for which with-replacement sampling outperforms without-replacement sampling, contrary to folklore.
- In Section 4, we prove that for non-trivial matrix ensembles (1) always holds with strict inequality for sufficiently small step sizes. Thus, for algorithms with a slowly decreasing step, without-replacement sampling always outperforms with-replacement sampling. On the other hand, we show that when optimal step sizes are chosen separately for with- and without-replacement sampling (but may not decrease to zero), with-replacement sampling can still perform better.
- In Section 5, we give an example convex learning task for which SGD using with-replacement sampling converges asymptotically faster than random-reshuffling.

## 1.1 Notation

In this paper, $\|\cdot\|$ of a vector always denotes the Euclidean $\ell_2$ norm, and $\|\cdot\|$ of an operator denotes the $\ell_2$ induced norm. We have $\mathbf{1}$ denote the all-1s vector. We let $\otimes$ denote the Kronecker product and $\oplus$ denote the matrix direct sum, such that $x \oplus y$ is the block diagonal matrix $\left[ \begin{smallmatrix} x & 0 \\ 0 & y \end{smallmatrix} \right]$. We let $\mathcal{S}_d$ denote

the set of symmetric $d \times d$ matrices over $\mathbb{R}$, and let $\mathcal{P}_d$ denote the set of symmetric positive definite $d \times d$ matrices. We let $\preceq$ denote inequality with respect to the positive definite ordering (i.e. $A \preceq B$ when $B - A \in \mathcal{P}_d$). When $\mathcal{K} \subset \mathbb{R}^d$ is a convex cone (a set closed under sums and non-negative scalar multiplication), and $x, y \in \mathbb{R}^d$, we say $x \preceq_\mathcal{K} y$ if and only if $y - x \in \mathcal{K}$, and for $A, B \in \mathbb{R}^{d \times d}$. We let $M(\mathcal{K})$ denote the set $\{A \in \mathcal{S}_d \mid \forall x \in \mathcal{K}, \ Ax \in \mathcal{K}\}$ of matrices that preserve the convex cone $\mathcal{K}$, and we note that $M(\mathcal{K})$ is also a convex cone. For brevity, we let $\mathsf{rr}_k : \mathcal{S}_d \times \mathcal{S}_d \times \cdots \times \mathcal{S}_d \to \mathcal{S}_d$ denote the "random reshuffling" function

$$\mathsf{rr}_k(A_1, A_2, \ldots, A_n) = \tfrac{1}{n!} \sum_{\sigma \in \mathcal{P}(n)} \prod_{i=1}^k A_{\sigma(i)},$$

and define $\mathsf{srr}_k$ similarly as the "symmetric random reshuffling" function

$$\mathsf{srr}_k(A_1, A_2, \ldots, A_n) = \tfrac{1}{n!} \sum_{\sigma \in \mathcal{P}(n)} \left( \prod_{i=1}^k A_{\sigma(i)} \right)^T \left( \prod_{i=1}^k A_{\sigma(i)} \right).$$

Note that this lets us write (1) more compactly as $\|\mathsf{rr}_n(A_1, \ldots)\| \leq \|\mathsf{rr}_1(A_1, \ldots)\|^n$.

## 1.2 Related Work

Conjecture 1 was a generalization of a line of older work on matrix arithmetic-mean geometric-mean inequalities for two matrices [4, 5]. It was proved by Recht and Ré [19] in the case of $n = 2$ and by Zhang [29] for $n = 3$. Duchi [7] proposed a variant, Conjecture 2, in which the sum appears outside the norm and proved it for $n = 2$, and it was extended to the $n = 3$ case by Israel et al. [12]. Albar et al. [2] proves a version of the inequality of Conjecture 1 that is weaker by a constant. Albar et al. [3] provides a non-constructive disproof of (2), and very recently Lai and Lim [14] gave a non-constructive disproof of (1) via a transformation to the noncommutative Positivstellensatz. Alaifari et al. [1] studies a related class of matrix rearrangement inequalities.

Several prior works have studied SGD on "noisy" learning problems, for which at the optimum $w^*$ it is not the case that $\nabla f_i(w^*) = 0$ for every component loss function $f_i$. Gürbüzbalaban et al. [8] exhibited an SGD variant for which random reshuffling converges at a $\Theta(1/t^2)$ rate, which improves on the $\Omega(1/t)$ rate of standard with-replacement-sampled SGD; similar results were also proved for the "noisy" case in other settings [10, 20, 22] Ying et al. [25] and Ying et al. [26] show that random reshuffling converges for variance-reduced algorithms, and Ying et al. [27] analyzes random reshuffling in the constant-step-size case. Meng et al. [15] studies a distributed variant of SGD with random shuffling, albeit one different from the one we study here (Algorithm 1). Beyond SGD, Oswald and Zhou [17] analyzes random reshuffling for methods such as Gauss-Seidel and Kaczmarz, and He et al. [11] studies scan order for Gibbs sampling.

## 2 Constructing a Counterexample

We start by outlining the main idea that underlies our counterexample. Fix some dimension $d \in \mathbb{N}$, and let $n = d$. For any permutation $\sigma \in \mathcal{P}(n)$, let $P_\sigma$ denote the *permutation matrix* over $\mathbb{R}^n$, such that $(P_\sigma x)_i = x_{\sigma(i)}$ for any vector $x \in \mathbb{R}^n$. The main idea is to construct a sequence of matrices $A_1, A_2, \ldots, A_n$ such that $P_\sigma A_i P_\sigma^T = A_{\sigma(i)}$ for any $\sigma$. For any permutation matrix $P_\varsigma$,

$$\mathsf{rr}_k(A_1, A_2, \ldots, A_n) = \mathsf{rr}_k(P_\varsigma A_1 P_\varsigma^T, P_\varsigma A_2 P_\varsigma^T, \ldots, P_\varsigma A_n P_\varsigma^T) = \frac{1}{n!} \sum_{\sigma \in \mathcal{P}(n)} \prod_{i=1}^k \left( P_\varsigma A_{\sigma(i)} P_\varsigma^T \right)$$

$$= P_\varsigma \left( \tfrac{1}{n!} \sum_{\sigma \in \mathcal{P}(n)} \prod_{i=1}^k A_{\sigma(i)} \right) P_\varsigma^T = P_\varsigma \left( \mathsf{rr}_k(A_1, A_2, \ldots, A_n) \right) P_\varsigma^T,$$

where the first equality holds because $\mathsf{rr}_k$ is a symmetric function, and the fourth holds because $P_\varsigma^T P_\varsigma = I$. This shows that $\mathsf{rr}_k(A_1, \ldots)$ is preserved by any permutation of its coordinates, and the only such matrices are of the form $X = \alpha \mathbf{1} \mathbf{1}^T + \beta I$, where $\mathbf{1}$ denotes the all-1s vector. With careful choice of the $A_i$, we can find formulas for $\alpha$ and $\beta$, and show that they violate Conjecture 1.

### 2.1 A Counterexample for the First Inequality

Define a family of vectors $y_k$ for $k \in \{1, \ldots, n\}$ such that for $i \neq k$ we have

$$(y_k)_k = \frac{\sqrt{n-1}}{n} \qquad \text{and} \qquad (y_k)_i = \frac{-1}{n \cdot \sqrt{n-1}}.$$

Consider the matrices $A_k = I + \mathbf{1}y_k^T + y_k\mathbf{1}^T$. For example, when $n = 5$, the matrices look like

$$A_1 = \begin{bmatrix} 1.8 & 0.3 & 0.3 & 0.3 & 0.3 \\ 0.3 & 0.8 & -0.2 & -0.2 & -0.2 \\ 0.3 & -0.2 & 0.8 & -0.2 & -0.2 \\ 0.3 & -0.2 & -0.2 & 0.8 & -0.2 \\ 0.3 & -0.2 & -0.2 & -0.2 & 0.8 \end{bmatrix}, \quad A_2 = \begin{bmatrix} 0.8 & 0.3 & -0.2 & -0.2 & -0.2 \\ 0.3 & 1.8 & 0.3 & 0.3 & 0.3 \\ -0.2 & 0.3 & 0.8 & -0.2 & -0.2 \\ -0.2 & 0.3 & -0.2 & 0.8 & -0.2 \\ -0.2 & 0.3 & -0.2 & -0.2 & 0.8 \end{bmatrix}, \quad \ldots$$

It is clear by construction that for any $\sigma \in \mathcal{P}(n)$, then $P_\sigma^T A_k P_\sigma = A_{\sigma(k)}$. It is also easily seen that the sum of the $y_k$ is zero, from which we can see immediately that

$$\mathsf{rr}_1(A_1, \ldots, A_n) = \frac{1}{n}\sum_{i=1}^{n} A_i = I + \mathbf{1}\left(\frac{1}{n}\sum_{i=1}^{n} y_i\right)^T + \left(\frac{1}{n}\sum_{i=1}^{n} y_i\right)\mathbf{1}^T = I.$$

So, $\|\mathsf{rr}_1(A_1, \ldots, A_n)\| = 1$. It is less immediate but still straightforward to show the following.

**Statement 1.** *If we define $\lambda$ (for any $k \in \mathbb{N}$) as*

$$\lambda = \left(1 + \frac{1}{n-1}\right)^{k/2} \cdot \cos\left(k \cdot \arcsin\left(\frac{1}{\sqrt{n}}\right)\right),$$

*then the random-reshuffled product of these matrices can be written as*

$$\mathsf{rr}_k(A_1, \ldots, A_n) = \lambda \cdot \frac{\mathbf{1}\mathbf{1}^T}{n} + \left(\frac{\lambda - 1}{n-1} + 1\right)\left(I - \frac{\mathbf{1}\mathbf{1}^T}{n}\right)$$

*and so $\lambda$ will be an eigenvalue of $\mathsf{rr}_n(A_1, \ldots, A_n)$ with corresponding eigenvector $\mathbf{1}$.*

We include a full derivation of this result—which is relatively easy to derive by hand—in the appendix. It is easy to find $n$ for which $|\lambda|$ is greater than 1. The smallest such $n$ is $n = 5$, where

$$\mathsf{rr}_n(A_1, \ldots, A_n) = \frac{29}{64} \cdot \left(I - \frac{\mathbf{1}\mathbf{1}^T}{5}\right) - \frac{19}{16} \cdot \frac{\mathbf{1}\mathbf{1}^T}{5}, \quad \text{and} \quad \lambda = \frac{-19}{16}.$$

This fact can be easily verified numerically, by computing $\mathsf{rr}_n$ directly for $n = 5$. Note that we can also have $\lambda > 1$, e.g. for $n = 40$, $\lambda \approx 1.655$. This shows directly that (1) is false. Note that while this setup may seem to suggest that a counterexample requires $n = d$ (while usually in linear regression $n \gg d$), it is straightforward to construct examples for which $n$ and $d$ are arbitrary (but no less than 5) by either adding additional $I$ matrices to the ensemble or adding additional dimensions containing only a 1 on the diagonal: this will change the norms of neither the arithmetic nor the geometric mean.

## 2.2 A Counterexample for the Second Inequality

Using our construction from the previous section, define a collection of positive semidefinite symmetric matrices $B_i$ such that $B_i^2 = A_i$. For these matrices, $\mathsf{srr}_1(B_1, \ldots, B_n) = \frac{1}{n}\sum_{i=1}^{n} B_i^2 = I$, so by induction

$$\frac{1}{n^n} \sum_{(s_1, \ldots, s_n) \in \{1, \ldots, n\}^n} \left(\prod_{i=1}^{n} B_{s_i}\right)^T \left(\prod_{i=1}^{n} B_{s_i}\right) = I.$$

Thus, its norm will be 1. Just as before, these matrices have the property that $P_\sigma B_i P_\sigma^T = B_{\sigma(i)}$ for any $\sigma$, so $P_\sigma \mathsf{srr}_n(B_1, \ldots) P_\sigma^T = \mathsf{srr}_n(B_1, \ldots)$ and the symmetrized random-reshuffled product can also be written as $\alpha \mathbf{1}\mathbf{1}^T + \beta I$. It is possible to perform the same sort of analysis as done in Section 2.1 to find an expression for the eigenvalues of $\mathsf{srr}_n(B_1, \ldots)$ explicitly as an analytic expression in $n$; however, since it is much more complicated and does not deliver additional insight, for lack of space we will just state the result for the particular case of $n = 10$. This case is convenient because $10 - 1 = 3^2$, and so $A_i$ is rational and thus $B_i$ is over $\mathbb{Q}(\sqrt{2})$, and so we can do exact arithmetic easily. In this case, the eigenvalue of $\mathsf{srr}_n(B_1, \ldots, B_n)$ with corresponding eigenvector $\mathbf{1}$ is exactly

$$\lambda = \frac{16623165607286458}{16677181699666569} + \frac{2195717144015980}{16677181699666569}\sqrt{2} \approx 1.183,$$

which shows directly that (2) is false, because if it were true this number could be at most 1.

## 2.3 A Counterexample for the Third Inequality

We can construct a counterexample to Conjecture 2 based on the "tight frame" example of Recht and Ré [19]. The "tight frame" example for $n = 2$ consists of symmetric projection matrices $A_k \in \mathbb{R}^{2 \times 2}$ for $k \in \{1, \dots, n\}$ defined as $A_k = u_k u_k^T$, where $u_k = \left[\cos\left(\frac{\pi k}{n}\right) \quad \sin\left(\frac{\pi k}{n}\right)\right]^T$. These matrices have the interesting property that their fixed order product $A_1 \cdot A_2 \cdots A_n$ has an asymptotically larger norm than the $n$th power of their mean, and they are used by Recht and Ré [19] to motivate why symmetrizing the order (by sampling without replacement rather than just going with some arbitrary fixed order) is important.

Starting with this, we construct the family of matrices $B_k$ defined by $B_k = \bigoplus_{\varsigma \in \mathcal{P}(n)} A_{\varsigma(k)}$, where $\bigoplus$ here denotes an indexed matrix direct sum (which constructs a block diagonal matrix). The direct sum has the important properties that $\|X \oplus Y\| = \max(\|X\|, \|Y\|)$ and that (if the dimensions match) $(X_1 \oplus X_2) \cdot (Y_1 \oplus Y_2) = (X_1 Y_1) \oplus (X_2 Y_2)$. As a consequence, for any permutation $\sigma$,

$$
\left\|\textstyle\prod_{i=1}^n B_{\sigma(i)}\right\| = \left\|\textstyle\prod_{i=1}^n \left(\bigoplus_{\varsigma \in \mathcal{P}(n)} A_{\varsigma(\sigma(i))}\right)\right\| = \left\|\bigoplus_{\varsigma \in \mathcal{P}(n)} \left(\textstyle\prod_{i=1}^n A_{\varsigma(\sigma(i))}\right)\right\|
$$
$$
= \max_{\varsigma \in \mathcal{P}(n)} \left\|\textstyle\prod_{i=1}^n A_{\varsigma(\sigma(i))}\right\| = \max_{\varsigma \in \mathcal{P}(n)} \left\|\textstyle\prod_{i=1}^n A_{\varsigma(i)}\right\| = \left\|\textstyle\prod_{i=1}^n A_i\right\|,
$$

where the last equality is a known property of the tight frame example, and the other equalities follow from properties of the direct sum. This means that all the terms on the left side of (3) for this example will be the same, and in particular

$$
\textstyle\frac{1}{n!} \sum_{\sigma \in \mathcal{P}(n)} \left\|\prod_{i=1}^n B_{\sigma(i)}\right\| = \left\|\prod_{i=1}^n A_i\right\|.
$$

By a similar argument, the right side of that equation will be

$$
\frac{1}{n^n} \sum_{f:\{1,\dots,n\}^n} \left\|\prod_{i=1}^n B_{f(i)}\right\| = \frac{1}{n^n} \sum_{f:\{1,\dots,n\}^n} \max_{\varsigma \in \mathcal{P}(n)} \left\|\prod_{i=1}^n A_{\varsigma(f(i))}\right\|.
$$

These formulas make it straightforward to compute these values, even though the matrices in question are of dimension $2 \cdot n!$. For the particular case of $n = 6$,

$$
\frac{1}{n!} \sum_{\sigma \in \mathcal{P}(n)} \left\|\prod_{i=1}^n B_{\sigma(i)}\right\| = \frac{9}{32}\sqrt{3} \approx 0.487 \quad \text{and} \quad \frac{1}{n^n} \sum_{f:\{1,\dots,n\}^n} \left\|\prod_{i=1}^n B_{f(i)}\right\| = \frac{26761}{124416} + \frac{29965}{248832}\sqrt{3} \approx 0.424.
$$

This is a counterexample to Conjecture 2.

## 3 A Machine Learning Example

Stochastic gradient descent is perhaps the central example in ML of an algorithm where sample order can affect convergence. Consider the parallel SGD algorithm described in Algorithm 1. Here, for every epoch, each of $M$ parallel workers runs $n$ iterations of SGD, using either with-replacement or without-replacement sampling. Then, the resulting weights are averaged among the workers to produce the starting value for the next epoch. This once-per-epoch averaging in some sense "simulates" the expected value in Conjecture 1, which makes Algorithm 1 a natural SGD-like algorithm to explore with the conjecture.[2] This is equivalent to the method of local SGD with periodic averaging [9, 28] with the averaging period equal to the epoch length. Based on folklore, random reshuffling should outperform standard sampling for this sort of algorithm. We will show that this is not always the case by constructing an example for which standard sampling converges at a linear rate, but random reshuffling fails to converge at all.

Consider the following matrix-completion-like task. We have an unknown rank-1 matrix $X \succeq 0$, and we are given noisy "measurements" from it of the form $u_i^T X v_i \approx a_i$ where we know $(u_i, v_i, a_i)$. We want to recover $X$ by solving the regularized least-squares minimization problem

$$
\text{minimize:} \quad \frac{1}{n} \sum_{i=1}^n \left(u_i^T X v_i - a_i\right)^2 + \frac{1}{2}\gamma \cdot \mathbf{tr}\left(X\right) \qquad \text{subject to } X \in \mathcal{P}_d, \ \mathrm{rank}(X) \leq 1.
$$

**Algorithm 1** Parallel SGD
---
1: **given:** $n$ loss functions $f_i$, step size scheme $\alpha_1, \alpha_2, \ldots$, initial state $w_0 \in \mathbb{R}^d$
2: **given:** number of epochs $K$, parallel machines $M$, replacement policy RP
3: **for** $k = 1$ **to** $K$ **do**
4:     **for all** $m \in \{1, \ldots, M\}$ **do in parallel on machine** $m$
5:         $u_{k,m,0} \leftarrow w_{k-1}$
6:         **if** $\mathrm{RP} = \texttt{with-replacement}$ **a.k.a** $\texttt{standard sampling}$ **then**
7:             **sample** $\sigma_{k,m}$ uniformly from the set of functions from $\{1, \ldots, n\}$ to $\{1, \ldots, n\}$
8:         **else if** $\mathrm{RP} = \texttt{without-replacement}$ **a.k.a** $\texttt{random reshuffling}$ **then**
9:             **sample** $\sigma_{k,m}$ uniformly from $\mathcal{P}(n)$
10:         **for** $t = 1$ **to** $n$ **do**
11:             $u_{k,m,t} \leftarrow u_{k,m,t-1} - \alpha_k \nabla f_{\sigma_{k,m}(t)} \left(u_{k,m,t-1}\right)$
12:     **average** $w_k \leftarrow \frac{1}{M} \sum_{m=1}^{M} u_{k,m,n}$
13: **return** $w_K$
---

To solve this more efficiently, we apply Algorithm 1 to a quadratic factorization [6] $X = yy^T$, a common technique which results in the equivalent unconstrained problem

$$\text{minimize: } f(y) = \frac{1}{n} \sum_{i=1}^{n} f_i(y) = \frac{1}{n} \sum_{i=1}^{n} \left(u_i^T yy^T v_i - a_i\right)^2 + \frac{1}{2}\gamma \|y\|^2 \qquad \text{subject to: } y \in \mathbb{R}^d.$$

We are now going to pick a particular dataset of $(u_i, v_i, a_i)$ such that the global optimum of $f$ is at $y = 0$, and where nearby $y = 0$, Algorithm 1 behaves like our counterexample of Section 2. To do this, notice that nearby $y = 0$,

$$
\begin{aligned}
I - \alpha \nabla f_i(y) &= \left((1 - \alpha\gamma I) - 2\alpha \left(u_i^T yy^T v_i - a_i\right)\left(v_i u_i^T + u_i v_i^T\right)\right) y \\
&= \left((1 - \alpha\gamma I) + 2\alpha a_i \left(v_i u_i^T + u_i v_i^T\right)\right) y + \mathcal{O}(y^3).
\end{aligned}
$$

So, if we choose $\alpha$, $\gamma$, $a_i$, $v_i$, and $u_i$ such that $(1 - \alpha\gamma I) + 2\alpha a_i \left(v_i u_i^T + u_i v_i^T\right) = (1 - \alpha\gamma)A_i$ where this $A_i$ is from our counterexample of Section 2, then when the algorithm is sufficiently close to $y = 0$, applying an iteration of SGD will behave like multiplying by a single matrix $A_i$, and averaging across multiple parallel workers will concentrate around the expected value over the sampled scan order. Concretely, we pick $n = 40$, a constant step size $\alpha = 0.1$, $\gamma = 0.05$, $M = 1000$, $K = 100$, $u_i = \mathbf{1}$, $v_i = y_i$ (the $y_i$ of Section 2), and $a_i = \frac{1-\alpha\gamma}{2\alpha}$; we initialize $w_0$ randomly such that $\|w_0\| = 1$. It is easy to see that the global optimum of this task is at $w^* = 0$, and it has no other stationary points. Note that this is not necessarily a very realistic setting (the number of parallel workers is large and the dataset is relatively small): the artificial setting is chosen to make the comparison stand out clearly. Running Algorithm 1 on this example produces the results shown in Figure 1. This shows empirically that, counter-intuitively, standard with-replacement random sampling can outperform random reshuffling for this algorithm.

## 4 Taking Step Size into Account

Many algorithms, including SGD, use a step size or learning rate that often decreases over time. Much of the previous work on random reshuffling has been done in such cases [8, 10, 20]. In this section, we develop a variant of Conjecture 1 that incorporates a step size, and prove that statement must hold true for sufficiently small step sizes. Our step-size-incorporating variant of Conjecture 1 is based on the following intuition. For a smooth objective, for $w$ close to the optimum $w^*$ where $\nabla f_i(w^*) = 0$,

$$w - \alpha \nabla f_i(w) \approx (I - \alpha \nabla^2 f_i(w))(w - w^*) + w^*;$$

for a quadratic function $f_i$, this approximation is exact. So we can model SGD with step size $\alpha$ by allowing the matrices $A_i$ in Conjecture 1 to vary as a function of a step size $\alpha$. We prove the following theorem about this modified inequality.

**Theorem 1** (Matrix AMGM Inequality, Sufficiently Small Step Size)**.** *Let $A_1, \ldots, A_n$ be a collection of continuously twice-differentiable functions from $\mathbb{R}_+$ to $\mathcal{S}_d$, that all satisfy $A_i(0) = I$ and that are non-trivial in the sense that they have no eigenvalue/eigenvector pairs shared among all the matrices*

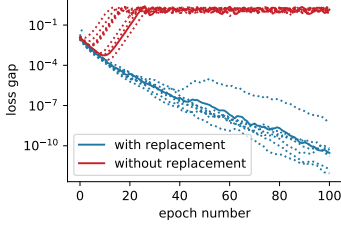 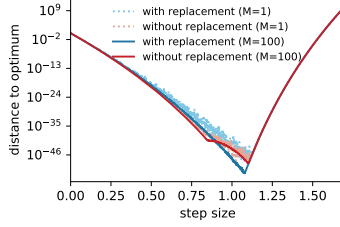 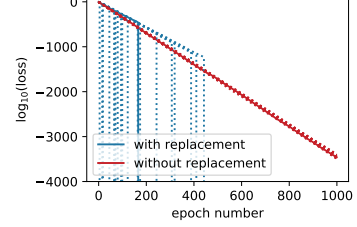

Figure 1: Loss gap of multiple random runs of Algorithm 1 on task of Section 3. Notice that random reshuffling fails to converge to the minimal loss.

Figure 2: Distance to optimum of Algorithm 1 for task of Section 4 after $K = 20$ epochs. Light dotted series indicate non-parallel SGD ($M = 1$).

Figure 3: Asymptotic convergence of SGD using with- and without-replacement sampling for multiple trials on the example of Section 5.

$A_1'(0), A_2'(0), \ldots, A_n'(0)$. *Then for any* $2 \le k \le n$, *there exists an* $\alpha_{\max} > 0$ *and a constant* $C > 0$ *such that such that for all* $0 < \alpha \le \alpha_{\max}$,

$$\|\mathsf{rr}_k(A_1(\alpha), A_2(\alpha), \ldots, A_n(\alpha))\| < \|\mathsf{rr}_1(A_1(\alpha), A_2(\alpha), \ldots, A_n(\alpha))\|^k - \alpha^2 C.$$

The proof of Theorem 1 is a straightforward combination of the following two lemmas. The main idea is that we can expand the $\mathsf{rr}_k$ expression approximately as a polynomial in $\alpha$, and then consider only the dominant quadratic $\alpha^2$ term, which we can bound directly. We defer the proof of the theorem and both lemmas to the appendix.

**Lemma 1** (Binomial Theorem for Random Reshuffling). *For any symmetric matrices* $X_1, \ldots, X_n$, *and any constants* $\alpha$ *and* $\beta$,

$$\mathsf{rr}_k(\alpha I + \beta X_1, \alpha I + \beta X_2, \ldots, \alpha I + \beta X_n) = \sum_{i=0}^{k} \binom{k}{i} \alpha^{k-i} \beta^k \mathsf{rr}_i(X_1, X_2, \ldots, X_n).$$

**Lemma 2.** *For any symmetric matrices* $X_1, \ldots, X_n \in \mathbb{R}^d$, *and for any* $u \in \mathbb{R}^d$ *such that* $\|u\| = 1$,

$$u^T (\mathsf{rr}_2(X_1, \ldots, X_n)) u \le \|\mathsf{rr}_1(X_1, \ldots, X_n)\|^2$$

*and equality can hold only if there exists a* $\lambda \in \mathbb{R}$ *such that for all* $i \in \{1, \ldots, n\}$, *u is an eigenvector of* $X_i$ *with eigenvalue* $\lambda$, *that is* $X_i u = \lambda u$.

We can use Theorem 1 to show that random reshuffling must outperform standard with-replacement sampling for slowly decaying step size schemes for noiseless convex quadratic problems, which can be thought of as a simplified model for optimization problems satisfying the strong growth condition of Schmidt and Roux [21]. Specifically, we study the following simplified class of problems.

**Definition 1.** We say that a set of loss functions $f_1, \ldots, f_n$ is a *noiseless convex quadratic problem* if the following conditions hold.

- **(Noiselessness.)** There exists a unique $w^* \in \mathbb{R}^d$ such that for all $i$, $\nabla f_i(w^*) = 0$.
- **(Convex quadratics.)** Each loss function $f_i$ is a convex quadratic $f_i(w) = w^T H_i w/2 + b_i^T w$.
- **(Lipschitz gradients.)** For some $L$, each loss function $f_i$ satisfies $\nabla^2 f_i(w) \preceq LI$.

Additionally, we say that the problem is *non-trivial* if the Hessian matrices $H_i = \nabla^2 f_i(w^*)$ share no eigenvalue/eigenvector pairs, i.e. there is no $(\lambda, u)$ with $u \ne 0$ such that for all $i$, $H_i u = \lambda u$.

Note that the last condition here is designed to rule out trivial cases such as all the loss functions $f_i$ being the same. Such trivial cases only happen on a set of measure 0 within the space of all possible loss functions $f_1, \ldots, f_n$ and initializations, so it is reasonable for us to exclude them. The class of problems described by Definition 1 includes the setting of the Randomized Kaczmarz method originally studied in Recht and Ré [19], as well as well-known tasks such as linear regression and ridge regression.

Theorem 1 now directly implies the following useful corollary, which says that without-replacement sampling outperforms with-replacement sampling whenever a diminishing but not square-summable step size scheme is used—including for the important case of $M = 1$ in Algorithm 1, which corresponds to the most common case of ordinary single-worker SGD.[3]

**Corollary 1.** *Consider Algorithm 1 on a non-trivial noiseless convex quadratic (using any $M$). Suppose that the step size scheme satisfies $0 < \alpha_i L < 1$ and is diminishing but not square-summable, i.e. $\lim_{k\to\infty} \alpha_k = 0$ and $\sum_{k=1}^{\infty} \alpha_k^2 = \infty$. Then for almost all initial values $w_0 \neq w^*$,*

$$\lim_{k\to\infty} \frac{\left\| \mathbf{E}[w_{k,\textit{without-replacement}}] - w^* \right\|}{\left\| \mathbf{E}[w_{k,\textit{with-replacement}}] - w^* \right\|} = 0.$$

Although it seems that Theorem 1 and Corollary 1 suggests that random reshuffling *is* indeed better when we allow the use of small step sizes, it has a significant limitation that would make that conclusion invalid: Corollary 1 only compares with-replacement and without-replacement sampling *using the same learning rate scheme for both*. Instead, when comparing two algorithms in the most fair way, we should select the best learning rate scheme for each algorithm individually. Surprisingly, when we allow this, we can give an example of a convex learning task for which with-replacement sampling, with a particular constant step size, converges faster than without-replacement sampling no matter what fixed step size scheme it uses.

The convex functions in question can be constructed directly from our counterexample of Section 2. Let $f_i(w) = \frac{1}{2} w^T H_i w$, where $H_i = \left(I - \frac{1}{2}A_i\right) \oplus \frac{1}{2} \oplus \frac{3}{2}$, where $\oplus$ denotes the matrix direct sum (such that $X \oplus Y$ is a block diagonal matrix with diagonal blocks $X$ and $Y$). This function must be convex because $H_i$ is positive semidefinite (which follows from the fact that the eigenvalues of $A_i$ are 0, 1, and 2). The main idea of this construction is to "force" the step size to be $\alpha = 1$, since otherwise either the $1/2$ or $3/2$ coordinate will end up converging at a suboptimal rate. Although a theoretical analysis of this would be straightforward, as such analysis delivers no new insight, we only validate that this example works empirically on a concrete example. We pick $n = 8$, $M = 100$, and $K = 20$, and we initialize $w_0$ from a Gaussian with less power on the last two coordinates, which makes the effect more visible. In Figure 2 we plot the distance to optimum after $K$ epochs for all step sizes $\alpha \in \{0, 0.001, 0.002, \ldots, 1.7\}$. Observe that while for smaller step sizes, random-reshuffling outperforms standard sampling—which validates Theorem 1—the best convergence overall is achieved by standard with-replacement sampling. Interestingly, the averaging of Algorithm 1 is necessary for this effect to happen for this task: in Figure 2 we also display results for standard SGD on the same problem (equivalent to setting $M = 1$) for which without-replacement sampling seems to consistently outperform with-replacement sampling.

## 5 Random Reshuffling Can be Worse Asymptotically Even for SGD

While we have shown Conjecture 1 is false, this does not necessarily imply random reshuffling can be worse with stochastic gradient descent, because the averaging present in Conjecture 1 is not present in plain SGD. Our counterexamples so far do *not* show random reshuffling performing worse with no averaging (Figure 2), so it remains consistent with our observations so far that random reshuffling *could* always outperform with-replacement sampling for SGD. But is this necessarily true?

In this section we will show that it is not: even for SGD without any averaging (Algorithm 1 with $m = 1$), we can construct a learning task for which with-replacement sampling converges strictly faster than random reshuffling—albeit one not based on a counterexample to any of the conjectures we have studied. Here, when we say it converges "strictly faster," we mean that for any coupling of the two algorithms, with-replacement sampling almost surely eventually achieves lower loss than random reshuffling and its loss remains lower for all time. The main idea behind this construction, which is based on the idea that any rank-deficient square matrix can be written as the product of three symmetric positive semidefinite matrices [24], is as follows. Consider the matrices

$$A_1 = \frac{1}{4}\begin{bmatrix} 2 & -1 & 1 \\ -1 & 2 & -1 \\ 1 & -1 & 1 \end{bmatrix}, \ A_2 = \frac{1}{4}\begin{bmatrix} 2 & 0 & -1 \\ 0 & 1 & 1 \\ -1 & 1 & 2 \end{bmatrix}, \ A_3 = \frac{1}{4}\begin{bmatrix} 0 & 0 & 0 \\ 0 & 1 & -1 \\ 0 & -1 & 2 \end{bmatrix}, \ \text{and } R = \frac{uu^T}{6} \text{ where } u = \begin{bmatrix} 1 \\ -2 \\ -1 \end{bmatrix}.$$

It is easy to check that all these matrices are symmetric and positive semidefinite (and $\preceq I$), and that $(A_1 A_2 A_3)^3 = 0$. Now, consider the behavior of SGD with step size $\alpha = 1/2$ on the problem where

$$f_1(w) = w^T(I - A_1)w, \ \ f_2(w) = w^T(I - A_2)w, \ \ f_3(w) = w^T(I - A_3)w, \ \ f_4(w) = w^T(I - R)w.$$

Observe that all these functions are convex, and that choosing to do a step with $f_1$ has the effect of multiplying $w$ by $A_1$, etc. This means that, for with-replacement sampling, if we sample our examples in the order $(1, 2, 3, 1, 2, 3, 1, 2, 3)$, the result after running those SGD steps will be to have

$w = 0$, *regardless of what value of $w$ we started with* (because $(A_1 A_2 A_3)^3 = 0$). Since we are guaranteed to sample that run of examples eventually, it follows that almost surely, after some finite amount of time with-replacement SGD achieves $w_t = 0 = w^*$, which minimizes the loss.

On the other hand, this sequence of samples can not occur for sampling without replacement. Instead, every epoch of SGD will contain an $R$, which "disrupts" the sequence. The sequence of matrices that are multiplied by $w$ will consist of sequences of $A_i$ matrices of length no more than 6 broken up by $R$ matrices. Because of the structure of $R$, for any matrix $X$, $6RXR = R \cdot u^T X u$: this means that the sequences of matrices $RA_\star \cdots A_\star R$ will reduce to the product of scalars $u^T A_\star \cdots A_\star u$. It is straightforward to verify numerically that this scalar is nonzero for any sequence of $A_\star$ matrices that can occur for sampling without replacement. So, for almost all initializations $w_0$, SGD with random reshuffling on this task *will never reach 0*, while SGD using with-replacement sampling is guaranteed to reach 0 in finite time. We conclude that with-replacement sampling converges asymptotically faster than random reshuffling for this task and step size.

In fact, we can say something even more general: among all step sizes $\alpha$ that satisfy $\alpha L \leq 1$, where $L$ is the smallest constant such that each $f_i$ is has $L$-Lipschitz gradients, SGD with $\alpha = 1/2$ using with-replacement sampling converges asymptotically faster than all other settings. That is, here with-replacement sampling is still converging asymptotically faster, even if we choose the optimal learning rates for both sampling strategies separately. We can see that this holds immediately from the fact that for $0 < \alpha < 1$, taking a step with respect to $f_i$ has the effect of multiplying $w$ by a full-rank matrix; such a step can never reach 0.

We explore this task empirically in Figure 3, where we ran a thousand epochs of SGD using both with- and without-replacement sampling on the example task we constructed in this section. Observe that for every run (we ran 20 independent runs), with-replacement sampling starts off a little worse, but quickly moves to zero loss, while without-replacement sampling continues to have nonzero loss for all time. Note that we needed to use exact arithmetic and look at points *very* close to the optimum for this effect to be observable: the figure ranges down to losses of $10^{-4000}$. Although not practical, this experiment does conclusively illustrate empirically that it is possible for with-replacement SGD to converge asymptotically faster than random-reshuffling, even when no averaging is used and when step sizes are chosen optimally for both algorithms.[4] In future work, it may be interesting to study whether and to what extent this sort of effect can occur in real learning tasks.

# 6   Conclusion

In this paper, we compared random reshuffling to with-replacement sampling for stochastic learning algorithms on noiseless problems. We found a counterexample to two longstanding conjectures from the literature (Conjectures 1 and 2) that would have implied random reshuffling is always no worse for many learning algorithms. Using this counterexample, we constructed concrete learning tasks for which with-replacement sampling outperforms without-replacement sampling in a way that can be observed empirically, even when step size is allowed to vary. This shows that, contrary to folklore, random-reshuffling *can* actually cause learning algorithms to converge asymptotically slower than with-replacement sampling on a particular problem. New insights will be required to develop theory that gets around our counterexamples to explain why random-reshuffling appears to consistently perform better on individual tasks in practice—even for noiseless problems. We hope that this work will bring us closer to a deeper understanding of the effect of scan order in machine learning.

## Broader Impact

We expect that the counterexamples presented in this work will have an impact on the ML theory community as we further try to understand the effect of scan order in large-scale optimization. We hope that these counterexamples will help guide future researchers towards proving more variants of Conjectures 1 and 2 that are true. Beyond this impact on the ML community, this work is primarily theoretical and does not present any foreseeable societal consequence.

## Funding Disclosure

No external funding (apart from usual faculty support by Cornell University) was used in support of this work. The author has an engagement with SambaNova Systems, but funding from SambaNova was not used to directly support this work.

## Footnotes

[1]Lai and Lim [14] can be considered parallel work to this paper.

[2]Note that the parallelism itself is not necessary here; what is necessary for our purposes is the *averaging*. The averaging is necessary (even for large n) because it models the expected value in the original inequality (1): without it the convergence rate may be effected by higher-order moments (not just the expected value). We study parallel SGD because it is a "real" method from the literature that uses averaging [9, 28].

[3]Note also that it should be straightforward to extend Corollary 1 to the case of "nice" convex losses, on the basis of the idea that any such functions must behave like quadratics locally in the neighborhood of $w^*$. But since this is not deliver any new insight, we do not present such a result here.

[4]Observe that this example could also be applied to a "noisy" dataset case with SVRG [13], showing that the phenomenon of random reshuffling sometimes being worse is limited neither to SGD nor to noisy datasets. Since the transformation is straightforward it is left as an exercise for the reader.

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
