[Supplementary Material]

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

# A    Additional Results

In this section, we present additional results that we believe are interesting, but that the main body of the paper was too short to contain.

## A.1    Violating the Conjecture by a Larger Margin

The expression in Section 2.1 makes it seem that our counterexample can violate Conjecture 1 only by at most a factor of $\exp(1/2)$, since this is an upper bound on the magnitude of $\lambda$. However, it is straightforward to use the counterexample to construct violations by an arbitrary factor. If we start with a sequence of $n$ matrices $A_1, \ldots, A_n \in \mathbb{R}^d$, we can use the Kronecker product to construct a sequence of $2n$ matrices in $\mathbb{R}^{d^2}$ as $A_1 \otimes I, A_2 \otimes I, \ldots, A_n \otimes I, I \otimes A_1, \ldots, I \otimes A_n$. In this case,

$$\mathsf{rr}_{2n}\left(A_1 \otimes I, A_2 \otimes I, \ldots, A_n \otimes I, I \otimes A_1, \ldots, I \otimes A_n\right) = \mathsf{rr}_n\left(A_1, \ldots, A_n\right) \otimes \mathsf{rr}_n\left(A_1, \ldots, A_n\right),$$

and so its norm will be

$$\left\|\mathsf{rr}_{2n}\left(A_1 \otimes I, A_2 \otimes I, \ldots, A_n \otimes I, I \otimes A_1, \ldots, I \otimes A_n\right)\right\| = \left\|\mathsf{rr}_n\left(A_1, \ldots, A_n\right)\right\|^2$$

and similarly for $\mathsf{srr}_{2n}(\cdots)$. On the other hand, the arithmetic means ($\mathsf{rr}_1$ and $\mathsf{srr}_1$, respectively) of these matrices will still be the identity matrix $I$. This sort of construction can be used to produce matrix ensembles that violate the inequalities of Conjecture 1 by arbitrarily large factors. Starting from the case of $n = 5$ of Section 2.1, we can show that for any $q \in \mathbb{N}$, there exists an ensemble of $n = 5q$ matrices $C_1, \ldots, C_n$ in dimension $d = 5^q$ such that

$$\left\|\mathsf{rr}_n(C_1, \ldots, C_n)\right\| = (19/16)^q \approx (1.035)^n \qquad \text{but} \qquad \mathsf{rr}_1(C_1, \ldots, C_n) = I.$$

Similarly, there exists an ensemble of $n = 10q$ matrices $C_1, \ldots, C_n$ in dimension $d = 10^q$ such that

$$\left\|\mathsf{srr}_n(C_1, \ldots, C_n)\right\| \approx (1.183)^q \approx (1.017)^n \qquad \text{but} \qquad \mathsf{srr}_1(C_1, \ldots, C_n) = I.$$

This places a non-trivial lower bound on any "loose" version of Conjecture 1. Note that Albar et al. [2] proved the dimension-free bound $\left\|\mathsf{rr}_n(A_1, \ldots, A_n)\right\| \leq n^n \cdot \left\|\mathsf{rr}_1(A_1, \ldots, A_n)\right\|^n$. Our lower bound here shows that this constant $n^n$ could only at best be improved to $O(1)^n$.

# B    Another Example: Markov chain Monte Carlo

In the main body of the text, we presented examples of learning tasks that were based on stochastic gradient descent. Another common class of ML algorithms where scan order matters is Markov chain Monte Carlo. Suppose that we have some unknown distribution $\pi$ over some sample space $\Omega$ we wish to sample from, and $P_1, P_2, \ldots, P_n$ are Markov transition operators each of which is reversible and has stationary distribution $\pi$ (but which are not necessarily ergodic). The classic example of this sort of setup is *Gibbs sampling*, in which each operator $P_i$ corresponds to the action of resampling a single variable from a joint distribution $\pi$ conditioned on the values of the other variables.

We consider two different ways of combining the individual transition operators into a single compound operator. *With-replacement* scan involves running $n$ inner iterations, where at each iteration the algorithm chooses a transition operator from $P_1, \ldots, P_n$ with replacement, then transitions the state according to that operator. We denote this compound operator $P_{\mathrm{with}}$. *Without-replacement* scan does the same thing, but samples without replacement. We denote this compound operator $P_{\mathrm{without}}$. Note that

$$P_{\mathrm{with}} = \left(\frac{1}{n}\sum_{i=1}^{n} P_i\right)^n \qquad \text{and} \qquad P_{\mathrm{without}} = \frac{1}{n!}\sum_{\sigma \in \mathcal{P}(n)}\prod_{i=1}^{n} P_{\sigma(i)},$$

from which we can see the connection to Conjecture 1.

It is natural to ask the question: how fast do these algorithms converge? The standard way to measure this is with the standard *total variation distance* from stationarity after $t$ steps, which is the maximum over all possible initial states of the total variation distance between the distribution of the Markov chain after $t$ steps and its stationary distribution. That is,

$$d(P, t) = \max_{x \in \Omega}\left\|P^t(x, \cdot) - \pi\right\|_{\mathrm{TV}} = \max_{x \in \Omega}\frac{1}{2}\sum_{y \in \Omega}\left|P^t(x, y) - \pi(y)\right|.$$

Figure 4: Distance-to-stationarity ratio of with-replacement versus without-replacement scan order for our constructed MCMC task. Notice that the without-replacement algorithm converges at a faster rate than the with-replacement algorithm for most values of $n$.

It would be intuitive to conjecture that it always holds (at least for sufficiently large $t$) that random reshuffling converges faster than standard with-replacement random sampling of the $P_i$, that is,

$$d\left(P_{\text{without}}, t\right) \leq d\left(P_{\text{with}}, t\right). \tag{4}$$

However, here we show that this is not necessarily always the case, by adapting our counterexample from Section 2. Consider the matrices

$$P_i = \frac{1}{2n \cdot (A_i)_{ii}} \cdot A_i \otimes \begin{bmatrix} 1 & -1 \\ -1 & 1 \end{bmatrix} + \frac{1}{2n} \cdot \mathbf{1}\mathbf{1}^T.$$

Notice that, since $(A_i)_{ii}$ is the largest-magnitude entry of $A_i$, this matrix is guaranteed to be non-negative, and since

$$P_i\mathbf{1} = \frac{1}{2n \cdot (A_i)_{ii}} \cdot (A_i\mathbf{1}) \otimes \left( \begin{bmatrix} 1 & -1 \\ -1 & 1 \end{bmatrix} \mathbf{1} \right) + \frac{1}{2n} \cdot \mathbf{1}\mathbf{1}^T\mathbf{1} = \frac{1}{2n \cdot (A_i)_{ii}} \cdot (A_i\mathbf{1}) \otimes (0) + \frac{1}{2n} \cdot \mathbf{1} \cdot 2n = \mathbf{1},$$

it is a stochastic matrix (and so it is a Markov operator over a state space of size $2n$). Additionally, since $P_i$ is symmetric (by construction, since the Kronecker product of symmetric matrices is symmetric), it follows that each $P_i$ must be reversible with stationary distribution $\pi = \frac{1}{2n}$. When $n - 1$ is a perfect square, $A_i$ is rational, and so it is straightforward to compute the compound operators $P_{\text{with}}$ and $P_{\text{without}}$ directly in exact arithmetic. We did this and computed the distances to stationarity $d\left(P_{\text{without}}, t\right)$ and $d\left(P_{\text{with}}, t\right)$ for $n \in \{5, 10, 17, 26, 37\}$ and for $T = 20$ total outer iterations. Figure 4 plots the ratio of these distances to stationarity. Notice that if (4) were true, we would expect these series to all lie above the dotted black line, at least for sufficiently large $t$. This shows that random reshuffling *can* converge more slowly than standard with-replacement random sampling, even for Markov chain Monte Carlo.

We should note here that our example is extremely artificial for MCMC, especially because both the with-replacement and without-replacement chains converge *very* fast. In practice even one iteration of either the with-replacement or without-replacement chains would produce estimates that are more than accurate enough. It is possible—even likely—that some analogue of the random-reshuffling-is-better conjecture holds for MCMC methods under more typical conditions, such as Gibbs sampling.

## C   Derivation of Section 2.1

In this section, we provide the derivation for Statement 1, the formula for the random-reshuffled product of our counterexample matrices.

We first state without proof the following facts about the vectors $y_i$, which can be easily checked.

$$\mathbf{1}^T y_i = 0,$$

$$y_i^T y_j = \frac{-1}{n(n-1)},$$

$$\sum_{i=1}^{n} y_i = 0, \quad \text{and}$$

$$\sum_{i \neq j} y_i y_j^T = -\sum_{i=1}^{n} y_i y_i^T = \frac{1}{n(n-1)} \mathbf{1}\mathbf{1}^T - \frac{1}{n-1} I.$$

Starting from the matrix family as defined in Section 2.1, for any permutation $\sigma$ and for any $k > 0$, the product of an even number of the permuted matrices $A_k - I$ can be written as

$$\prod_{i=1}^{2k} \left( A_{\sigma(i)} - I \right) = \prod_{i=1}^{2k} \left( \mathbf{1} y_i^T + y_i \mathbf{1}^T \right)$$

$$= \left( \frac{-1}{n(n-1)} \right)^k \cdot n^{k-1} \mathbf{1}\mathbf{1}^T + \left( \frac{-1}{n(n-1)} \right)^{k-1} n^k y_{\sigma(1)} y_{\sigma(2k)}^T$$

$$= \left( \frac{-1}{n-1} \right)^k \left( \frac{\mathbf{1}\mathbf{1}^T}{n} - n(n-1) \cdot y_{\sigma(1)} y_{\sigma(2k)}^T \right),$$

and similarly for an odd number of matrices

$$\prod_{i=1}^{2k-1} \left( A_{\sigma(i)} - I \right) = \left( \frac{-1}{n-1} \right)^{k-1} \left( \mathbf{1} y_{\sigma(2k-1)}^T + y_{\sigma(1)} \mathbf{1}^T \right)$$

because there are only two "paths" through the product since any $\mathbf{1}^T y_i = 0$. Therefore,

$$\mathrm{rr}_{2k-1}(A_1 - I, \ldots, A_n - I) = \frac{1}{n!} \sum_{\sigma \in \mathcal{P}(n)} \left( \frac{-1}{n-1} \right)^{k-1} \left( \mathbf{1} y_{\sigma(2k-1)}^T + y_{\sigma(1)} \mathbf{1}^T \right)$$

$$= \left( \frac{-1}{n-1} \right)^{k-1} \left( \mathbf{1} \left( \frac{1}{n} \sum_{i=1}^{n} y_i \right)^T + \left( \frac{1}{n} \sum_{i=1}^{n} y_i \right) \mathbf{1}^T \right) = 0,$$

and

$$\mathrm{rr}_{2k}(A_1 - I, \ldots, A_n - I) = \frac{1}{n!} \sum_{\sigma \in \mathcal{P}(n)} \left( \frac{-1}{n-1} \right)^k \left( \frac{\mathbf{1}\mathbf{1}^T}{n} - n(n-1) \cdot y_{\sigma(1)} y_{\sigma(2k)}^T \right)$$

$$= \frac{1}{n(n-1)} \sum_{i \neq j} \left( \frac{-1}{n-1} \right)^k \left( \frac{\mathbf{1}\mathbf{1}^T}{n} - n(n-1) \cdot y_i y_j^T \right)$$

$$= \left( \frac{-1}{n-1} \right)^k \left( \frac{\mathbf{1}\mathbf{1}^T}{n} - \sum_{i \neq j} y_i y_j^T \right)$$

$$= \left( \frac{-1}{n-1} \right)^k \left( \frac{1}{n-1} I + \frac{n-2}{n(n-1)} \mathbf{1}\mathbf{1}^T \right).$$

We can write these in a single expression as, for $k > 0$,

$$\mathrm{rr}_k(A_1 - I, \ldots, A_n - I) = \frac{1}{2} \left( \left( \frac{\mathrm{i}}{\sqrt{n-1}} \right)^k + \left( \frac{-\mathrm{i}}{\sqrt{n-1}} \right)^k \right) \left( \frac{1}{n-1} I + \frac{n-2}{n(n-1)} \mathbf{1}\mathbf{1}^T \right)$$

where $\mathrm{i}$ is the imaginary unit. If we define

$$C_n = \frac{1}{n-1} I + \frac{n-2}{n(n-1)} \mathbf{1}\mathbf{1}^T,$$

then by the binomial theorem, for any $m \in \mathbb{N}$, the original random-reshuffled product can be written as

$$\mathsf{rr}_m(A_1, \ldots, A_n) = I + \sum_{k=1}^{m} \binom{m}{k} \mathsf{rr}_k(A_1 - I, \ldots, A_n - I)$$

$$= I + \frac{1}{2} \cdot C_n \cdot \sum_{k=1}^{m} \binom{m}{k} \left( \left( \frac{\mathrm{i}}{\sqrt{n-1}} \right)^k + \left( \frac{-\mathrm{i}}{\sqrt{n-1}} \right)^k \right)$$

$$= I + \frac{1}{2} \cdot C_n \cdot \left( \left( 1 + \frac{\mathrm{i}}{\sqrt{n-1}} \right)^m - 1 + \left( 1 - \frac{\mathrm{i}}{\sqrt{n-1}} \right)^m - 1 \right)$$

$$= I + C_n \cdot \left( \left( 1 + \frac{1}{n-1} \right)^{m/2} \cdot \cos \left( m \cdot \arctan \left( \frac{1}{\sqrt{n-1}} \right) \right) - 1 \right).$$

In particular, since $C_n \mathbf{1} = \mathbf{1}$, it follows that one eigenvalue of this will be

$$\lambda = \left( 1 + \frac{1}{n-1} \right)^{m/2} \cdot \cos \left( m \cdot \arctan \left( \frac{1}{\sqrt{n-1}} \right) \right).$$

This is what we set out to prove. Also note that for large $n$, if $m = n$,

$$\lambda \approx \exp \left( \frac{1}{2} \right) \cdot \cos \left( \sqrt{n} \right),$$

so we should expect to find $n$ for which $|\lambda| > 1$ fairly easily.

# D  Derivation for Section 2.2

In this section, we provide derivation details for our claims in Section 2.2. First, we note that since $A_i$ has eigenvalues in $\{0, 1, 2\}$ with multiplicity 1 on the 0 and 2, the matrix $A_i(A_i - I)$ is just $2uu^T$ where $u$ is the unit eigenvector with eigenvalue 2. So, if $B_i^2 = A_i$,

$$A_i + \frac{\sqrt{2} - 2}{2} \cdot A_i(A_i - I) = B_i.$$

Now, recall that $B_i$ satisfies $P_\sigma B_i P_\sigma^T = B_{\sigma(i)}$. So,

$$\mathsf{srr}_n(B_1, \ldots, B_n) = \frac{1}{n!} \sum_{\sigma \in \mathcal{P}(n)} \left( \prod_{i=1}^{n} B_{\sigma(i)} \right)^T \left( \prod_{i=1}^{n} B_{\sigma(i)} \right)$$

$$= \frac{1}{n!} \sum_{\sigma \in \mathcal{P}(n)} \left( \prod_{i=1}^{n} P_\sigma B_i P_\sigma^T \right)^T \left( \prod_{i=1}^{n} P_\sigma B_i P_\sigma^T \right)$$

$$= \frac{1}{n!} \sum_{\sigma \in \mathcal{P}(n)} P_\sigma \left( \prod_{i=1}^{n} B_i \right)^T \left( \prod_{i=1}^{n} B_i \right) P_\sigma^T.$$

Now, since we know that $\mathbf{1}$ must be an eigenvector of $\mathsf{srr}_n(B_1, \ldots, B_n)$, it follows that the corresponding eigenvalue must be

$$\lambda = \frac{1}{\mathbf{1}^T \mathbf{1}} \cdot \mathbf{1}^T \mathsf{srr}_n(B_1, \ldots, B_n) \mathbf{1}$$

$$= \frac{1}{n} \cdot \frac{1}{n!} \sum_{\sigma \in \mathcal{P}(n)} \mathbf{1}^T P_\sigma \left( \prod_{i=1}^{n} B_i \right)^T \left( \prod_{i=1}^{n} B_i \right) P_\sigma^T \mathbf{1}$$

$$= \frac{1}{n} \cdot \frac{1}{n!} \sum_{\sigma \in \mathcal{P}(n)} \mathbf{1}^T \left( \prod_{i=1}^{n} B_i \right)^T \left( \prod_{i=1}^{n} B_i \right) \mathbf{1}$$

$$= \frac{1}{n} \cdot \mathbf{1}^T \left( \prod_{i=1}^{n} B_i \right)^T \left( \prod_{i=1}^{n} B_i \right) \mathbf{1}.$$

We can easily compute this directly to get the eigenvalue. Doing this produces the value given in the body of the paper. We can validate this with the following Julia code.

```julia
using LinearAlgebra

# define a type for numbers of the form [x + y * sqrt(2)]
struct QR2 <: Number
    x :: Rational{BigInt};
    y :: Rational{BigInt};
end

import Base: +, -, *, //, ==, zero, one, Float64, promote_rule
*(a::QR2, b::QR2) = QR2(a.x * b.x + 2 * a.y * b.y, a.x * b.y + a.y * b.x);
+(a::QR2, b::QR2) = QR2(a.x + b.x, a.y + b.y);
-(a::QR2, b::QR2) = QR2(a.x - b.x, a.y - b.y);
-(b::QR2) = QR2(-b.x, -b.y);
//(a::QR2, q) = QR2(a.x // BigInt(q), a.y // BigInt(q));
zero(::Type{QR2}) = QR2(0, 0);
one(::Type{QR2}) = QR2(1, 0);
zero(a::QR2) = QR2(0, 0);
one(a::QR2) = QR2(1, 0);
Float64(a::QR2) = Float64(BigFloat(a.x) + sqrt(BigFloat(2)) * BigFloat(a.y));
==(a::QR2, b::QR2) = (a.x == b.x) && (a.y == b.y);
QR2(x::Rational) = QR2(x, 0);
QR2(x::Int64) = QR2(BigInt(x) // 1, 0);
promote_rule(::Type{QR2}, ::Type{Integer}) = QR2
promote_rule(::Type{QR2}, ::Type{Int64}) = QR2
promote_rule(::Type{QR2}, ::Type{Rational}) = QR2
promote_rule(::Type{QR2}, ::Type{Rational{BigInt}}) = QR2

# the parameters for the example
n = 10;
sqrtn1 = BigInt(sqrt(n-1));

# define the A matrices
ys = [[(k == i) ? sqrtn1//n : -1//(n*sqrtn1) for i = 1:n] for k = 1:n];
As = [I + ones(BigInt,n) * y' + y * ones(BigInt,n)' for y in ys];

# define the B matrices
Bs = [A .+ ((QR2(0,1) - QR2(2,0))//2) * A * (A - I) for A in As];

# assert that, indeed, B^2 = A
for i = 1:n
        @assert(Bs[i]^2 == QR2.(As[i]));
end

# compute the eigenvalue
lambda = sum(transpose(prod(Bs)) * (prod(Bs))) // n
println(lambda);
println(Float64(lambda));

# assert that it is the value in the paper
@assert(lambda == QR2(16623165607286458//16677181699666569,
    ↪ 2195717144015980//16677181699666569));

# validate that this also works in floating point
Asfp = [Float64.(A) for A in As];
Bsfp = [A .+ ((sqrt(2) - 2)/2) * (A^2 - A) for A in Asfp];

# assert that, indeed, B^2 = A, up to some tolerance, and that B is PSD
tol = 1e-8;
for i = 1:n
        @assert(all(eigvals(Bsfp[i]) .>= -tol));
        @assert(maximum(Bsfp[i]^2 .- Asfp[i]) < tol);
```

```
61  end
62
63  # compute the floating-point estimate of the eigenvalue
64  lambdafp = sum(transpose(prod(Bsfp)) * (prod(Bsfp))) / n;
65  println(lambdafp);
66
67  # check that it's equal to the value we computed exactly
68  @assert(abs(lambdafp - Float64(lambda)) < tol);
```

## E   Code to Verify the Counterexample to Conjecture 2

Here, we include our Julia code to verify that the counterexample described in Section 2.3 is actually a counterexample to Conjecture 2. Our code here also re-computes the numeric constants given in that section for the norms of the various matrices.

```
1   using Statistics
2   using Combinatorics
3   using LinearAlgebra
4
5   # we first construct a number type to do arithmetic in the field of Q(sqrt(3))
6   struct QR3 <: Number
7       x :: Rational{BigInt};
8       y :: Rational{BigInt};
9   end
10
11  import Base.+
12  import Base.-
13  import Base.*
14  import Base.//
15  import Base.==
16  import Base.zero
17  import Base.conj
18  import Base.sqrt
19  import Base.one
20  import Base.isless
21  import Base.Float64
22  import Base.promote_rule
23
24  function *(a::QR3, b::QR3)
25      return QR3(a.x * b.x + 3 * a.y * b.y, a.x * b.y + a.y * b.x);
26  end
27
28  function +(a::QR3, b::QR3)
29      return QR3(a.x + b.x, a.y + b.y);
30  end
31
32  function -(a::QR3, b::QR3)
33      return QR3(a.x - b.x, a.y - b.y);
34  end
35
36  function -(b::QR3)
37      return QR3(-b.x, -b.y);
38  end
39
40  function //(a::QR3, q::I) where {I <: Integer}
41      return QR3(a.x // BigInt(q), a.y // BigInt(q));
42  end
43
44  function zero(::Type{QR3})
45      return QR3(0, 0);
46  end
47
```

```julia
48  function one(::Type{QR3})
49      return QR3(1, 0);
50  end
51
52  function zero(a::QR3)
53      return QR3(0, 0);
54  end
55
56  function conj(a::QR3)
57      return a;
58  end
59
60  function isless(a::QR3, b::QR3)
61          cx = a.x - b.x;
62          cy = a.y - b.y;
63          if (cx >= 0)&&(cy >= 0)
64                  return false;
65          elseif (cx < 0)&&(cy < 0)
66                  return true;
67          elseif (cx < 0)
68                  return cx^2 > 3 * cy^2;
69          elseif (cy < 0)
70                  return 3 * cy^2 > cx^2;
71          else
72                  error();
73          end
74  end
75
76  function one(a::QR3)
77      return QR3(1, 0);
78  end
79
80  function Float64(a::QR3)
81      return Float64(BigFloat(a.x) + sqrt(BigFloat(3)) * BigFloat(a.y));
82  end
83
84  function ==(a::QR3, b::QR3)
85          return (a.x == b.x) && (a.y == b.y);
86  end
87
88  function QR3(x::Rational)
89      return QR3(x, 0);
90  end
91
92  function QR3(x::Int64)
93      return QR3(BigInt(x) // 1, 0);
94  end
95
96  function Rational(a::QR3)
97          @assert(a.y == 0);
98          return a.x;
99  end
100
101 promote_rule(::Type{QR3}, ::Type{Integer}) = QR3
102 promote_rule(::Type{QR3}, ::Type{Int64}) = QR3
103 promote_rule(::Type{QR3}, ::Type{Rational}) = QR3
104 promote_rule(::Type{QR3}, ::Type{Rational{BigInt}}) = QR3
105
106 # a function to try to compute the square root of a rational number, or error
    ↪   otherwise
107 function sqrtq3(x::Rational{BigInt})
108     n = x.num * x.den;
109     if (sqrt(n) == floor(sqrt(n)))
110         return QR3(BigInt(sqrt(n))//x.den, 0);
111     elseif (sqrt(n/3) == floor(sqrt(n/3)))
```

```
112          return QR3(0, BigInt(sqrt(n/3)))//x.den);
113      else
114          error();
115      end
116  end
117
118  # every matrix we'll be computing the norm of is rank-1,
119  # so its l2 norm is equal to the square root of the trace of A*A'
120  function l2norm(A::Array{QR3,2})
121      return sqrtq3(Rational(tr(A*A')));
122  end
123
124  # define the counterexample dataset
125  n = 6;
126  us = [[QR3(0,1//2), QR3(1//2,0)], [QR3(1//2,0), QR3(0,1//2)], [QR3(0,0), QR3(1,0)],
      ↪  [QR3(-1//2,0), QR3(0,1//2)], [QR3(0,-1//2), QR3(1//2,0)], [QR3(-1,0),
      ↪  QR3(0,0)]];
127  As = [u*u' for u in us];
128
129
130  # on the left side of the inequality of Conjecture 2
131  left_expr = maximum(l2norm(prod(p)) for p in permutations(As));
132  println("left side = $left_expr = $(Float64(left_expr))")
133
134
135  # on the right side of the inequality of Conjecture 2
136  NX = [l2norm(As[i1]*As[i2]*As[i3]*As[i4]*As[i5]*As[i6]) for i1=1:n, i2=1:n, i3=1:n,
      ↪  i4=1:n, i5=1:n, i6=1:n];
137
138  RX = -1 * ones(QR3,6,6,6,6,6,6);
139  for i1=1:n, i2=1:n, i3=1:n, i4=1:n, i5=1:n, i6=1:n
140      if (RX[i1,i2,i3,i4,i5,i6] == -1)
141          lmax = maximum(NX[p[i1],p[i2],p[i3],p[i4],p[i5],p[i6]] for p in
              ↪  permutations(1:n));
142          for p in permutations(1:n)
143              RX[p[i1],p[i2],p[i3],p[i4],p[i5],p[i6]] = lmax;
144          end
145      end
146  end
147
148  right_expr = sum(RX)//length(RX);
149  println("right side = $right_expr = $(Float64(right_expr))")
```

# F   Proof of Theorem 1

The proof of Theorem 1 is, as stated in the main body of the paper, a straightforward combination of the following two lemmas. The main idea is that we can expand the $rr_k$ expression approximately as a polynomial in $\alpha$, and then consider only the dominant quadratic $\alpha^2$ term, which we can bound directly.

**Lemma 3** (Binomial Theorem for Random Reshuffling). *For any symmetric matrices $X_1, \ldots, X_n$, and any constants $\alpha$ and $\beta$,*

$$rr_k(\alpha I + \beta X_1, \alpha I + \beta X_2, \ldots, \alpha I + \beta X_n) = \sum_{i=0}^{k} \binom{k}{i} \alpha^{k-i} \beta^k rr_i(X_1, X_2, \ldots, X_n).$$

**Lemma 4.** *For any symmetric matrices $X_1, \ldots, X_n \in \mathbb{R}^d$, and for any $u \in \mathbb{R}^d$ such that $\|u\| = 1$,*

$$u^T \left( rr_2(X_1, \ldots, X_n) \right) u \le \|rr_1(X_1, \ldots, X_n)\|^2$$

*and equality can hold only if there exists a $\lambda \in \mathbb{R}$ such that for all $i \in \{1, \ldots, n\}$, $u$ is an eigenvector of $X_i$ with eigenvalue $\lambda$, that is $X_i u = \lambda u$.*

We start by proving the lemmas, then prove the theorem.

### F.1 Proofs of Lemmas

In this section, we prove the lemmas presented in Section 4. We start by proving the following lemma, which will be useful for proving Lemma 4.

**Lemma 5.** *Let $\omega$ be a primitive $n$th root of unity. Then, for any symmetric real matrices $X_1, \ldots, X_n$,*

$$\frac{1}{n \cdot (n-1)} \sum_{i=1}^{n} \sum_{j \neq i} X_i X_j = \left( \frac{1}{n} \sum_{i=1}^{n} X_i \right)^2 - \frac{1}{n!} \sum_{\sigma \in \mathcal{P}(n)} \left( \frac{1}{n} \sum_{i=1}^{n} \omega^{\sigma(i)} X_i \right)^* \left( \frac{1}{n} \sum_{i=1}^{n} \omega^{\sigma(i)} X_i \right).$$

*Proof.* It is clear that the expressions on both sides of this equation will contain only terms of the form $X_i X_j$, where possibly $i = j$. When $i = j$, the left side will have no terms in $X_i^2$, and the right side will have

$$\frac{1}{n^2} X_i^2 - \frac{1}{n!} \sum_{\sigma \in \mathcal{P}(n)} \left( \frac{1}{n} \omega^{\sigma(i)} X_i \right)^* \left( \frac{1}{n} \omega^{\sigma(i)} X_i \right) = \frac{1}{n^2} X_i^2 - \frac{1}{n!} \sum_{\sigma \in \mathcal{P}(n)} \left( \frac{1}{n^2} X_i^2 \right)$$

$$= \frac{1}{n^2} X_i^2 - \left( \frac{1}{n^2} X_i^2 \right)$$

$$= 0,$$

where this holds since $\left( \omega^k \right) \cdot \left( \omega^k \right)^* = 1$ because $\omega$ is a root of unity. This is what we want for the $i = j$ case.

On the other hand, for $i \neq j$, the left side will have just the term $\frac{1}{n \cdot (n-1)} \cdot X_i X_j$. On the right side, we have

$$\frac{1}{n^2} X_i X_j - \frac{1}{n!} \sum_{\sigma \in \mathcal{P}(n)} \left( \frac{1}{n} \omega^{\sigma(i)} X_i \right)^* \left( \frac{1}{n} \omega^{\sigma(j)} X_j \right)$$

$$= \frac{1}{n^2} X_i X_j - \frac{1}{n!} \sum_{\sigma \in \mathcal{P}(n)} \left( \frac{1}{n^2} \omega^{\sigma(j) - \sigma(i)} X_i X_j \right)$$

$$= \frac{1}{n^2} X_i X_j \left( 1 - \frac{1}{n!} \sum_{\sigma \in \mathcal{P}(n)} \omega^{\sigma(i) - \sigma(j)} \right)$$

$$= \frac{1}{n^2} X_i X_j \left( 1 - \frac{1}{n \cdot (n-1)} \sum_{i=1}^{n} \sum_{j \neq i} \omega^{j-i} \right)$$

$$= \frac{1}{n^2} X_i X_j \left( 1 - \frac{1}{n \cdot (n-1)} \left( \sum_{i=1}^{n} \sum_{j=1}^{n} \omega^{j-i} - \sum_{i=1}^{n} \omega^{i-i} \right) \right)$$

$$= \frac{1}{n^2} X_i X_j \left( 1 - \frac{1}{n \cdot (n-1)} \left( \left( \sum_{i=1}^{n} \omega^{-i} \right) \left( \sum_{j=1}^{n} \omega^{j} \right) - \sum_{i=1}^{n} 1 \right) \right)$$

$$= \frac{1}{n^2} X_i X_j \left( 1 - \frac{1}{n \cdot (n-1)} (0 - n) \right)$$

$$= \frac{1}{n^2} X_i X_j \left( 1 + \frac{1}{n-1} \right)$$

$$= \frac{1}{n \cdot (n-1)} X_i X_j.$$

This is the desired result. $\square$

Using this lemma, we can now prove Lemma 4.

**Lemma 4.** *For any symmetric matrices $X_1, \ldots, X_n \in \mathbb{R}^d$, and for any $u \in \mathbb{R}^d$ such that $\|u\| = 1$,*

$$u^T \left( \mathrm{rr}_2(X_1, \ldots, X_n) \right) u \leq \|\mathrm{rr}_1(X_1, \ldots, X_n)\|^2$$

*and equality can hold only if there exists a $\lambda \in \mathbb{R}$ such that for all $i \in \{1, \ldots, n\}$, $u$ is an eigenvector of $X_i$ with eigenvalue $\lambda$, that is $X_i u = \lambda u$.*

*Proof.* By Lemma 5,

$$u^T \left( \frac{1}{n \cdot (n-1)} \sum_{i=1}^{n} \sum_{j \neq i} X_i X_j \right) u = u^T \left( \frac{1}{n} \sum_{i=1}^{n} X_i \right)^2 u$$

$$- u^T \left( \frac{1}{n!} \sum_{\sigma \in \mathcal{P}(n)} \left( \frac{1}{n} \sum_{i=1}^{n} \omega^{\sigma(i)} X_i \right)^* \left( \frac{1}{n} \sum_{i=1}^{n} \omega^{\sigma(i)} X_i \right) \right) u.$$

Since each term in the rightmost part of this expression is of the form $A^* A$, it follows that the resulting quadratic form must be positive semidefinite, and so

$$u^T \left( \frac{1}{n!} \sum_{\sigma \in \mathcal{P}(n)} \left( \frac{1}{n} \sum_{i=1}^{n} \omega^{\sigma(i)} X_i \right)^* \left( \frac{1}{n} \sum_{i=1}^{n} \omega^{\sigma(i)} X_i \right) \right) u \geq 0.$$

So,

$$u^T \left( \frac{1}{n \cdot (n-1)} \sum_{i=1}^{n} \sum_{j \neq i} X_i X_j \right) u \leq u^T \left( \frac{1}{n} \sum_{i=1}^{n} X_i \right)^2 u.$$

The first part of the lemma follows from the simple observation that by definition

$$u^T \left( \frac{1}{n} \sum_{i=1}^{n} X_i \right)^2 u = \left\| \left( \frac{1}{n} \sum_{i=1}^{n} X_i \right) u \right\|^2 \leq \left\| \frac{1}{n} \sum_{i=1}^{n} X_i \right\|^2 \|u\|^2 = \left\| \frac{1}{n} \sum_{i=1}^{n} X_i \right\|^2.$$

To prove the second part of the theorem, notice that since we just showed that

$$u^T \left( \frac{1}{n \cdot (n-1)} \sum_{i=1}^{n} \sum_{j \neq i} X_i X_j \right) u \leq u^T \left( \frac{1}{n} \sum_{i=1}^{n} X_i \right)^2 u \leq \left\| \frac{1}{n} \sum_{i=1}^{n} X_i \right\|^2,$$

equality between the left and right terms of this statement can only hold if *both* stated inequalities hold with equality. It is well-known that for symmetric matrices $A$ and unit vectors $u$, $u^T A^2 u = \|A\|^2$ only if $u$ is an eigenvector of $A$. Thus, our right inequality will hold with equality only if there exists a $\lambda \in \mathbb{R}$ such that

$$\left( \frac{1}{n} \sum_{i=1}^{n} X_i \right) u = \lambda u.$$

On the other hand, by Lemma 1, the left inequality will hold with equality only when

$$u^T \left( \frac{1}{n!} \sum_{\sigma \in \mathcal{P}(n)} \left( \frac{1}{n} \sum_{i=1}^{n} \omega^{\sigma(i)} X_i \right)^* \left( \frac{1}{n} \sum_{i=1}^{n} \omega^{\sigma(i)} X_i \right) \right) u = 0.$$

This only happens when

$$\frac{1}{n!} \sum_{\sigma \in \mathcal{P}(n)} u^T \left( \frac{1}{n} \sum_{i=1}^{n} \omega^{\sigma(i)} X_i \right)^* \left( \frac{1}{n} \sum_{i=1}^{n} \omega^{\sigma(i)} X_i \right) u = 0.$$

Since each of the matrices of the form $AA^*$ produces a positive semidefinite quadratic form, it follows that for every permutation $\sigma$,

$$u^T \left( \frac{1}{n} \sum_{i=1}^{n} \omega^{\sigma(i)} X_i \right)^* \left( \frac{1}{n} \sum_{i=1}^{n} \omega^{\sigma(i)} X_i \right) u \geq 0.$$

But then, equality can be attained only when *all* of these terms are 0, that is

$$u^T \left( \frac{1}{n} \sum_{i=1}^n \omega^{\sigma(i)} X_i \right)^* \left( \frac{1}{n} \sum_{i=1}^n \omega^{\sigma(i)} X_i \right) u = 0 \qquad \text{for every } \sigma \in \mathcal{P}(n).$$

Since

$$u^T \left( \frac{1}{n} \sum_{i=1}^n \omega^{\sigma(i)} X_i \right)^* \left( \frac{1}{n} \sum_{i=1}^n \omega^{\sigma(i)} X_i \right) u = \left\| \left( \frac{1}{n} \sum_{i=1}^n \omega^{\sigma(i)} X_i \right) u \right\|^2,$$

it follows that this only holds when

$$\left( \frac{1}{n} \sum_{i=1}^n \omega^{\sigma(i)} X_i \right) u = 0 \qquad \text{for every } \sigma \in \mathcal{P}(n).$$

Since this holds for *any* permutation, it follows that it holds for any scaling vector in the span of the permutations. Explicitly,

$$\sum_{i=1}^n b_i X_i u = 0 \qquad \text{for every } b \in \text{span} \left( \begin{bmatrix} \omega^{\sigma(1)} \\ \vdots \\ \omega^{\sigma(n)} \end{bmatrix} \,\middle|\, \sigma \in \mathcal{P}(n) \right).$$

It is not hard to see that this span is just the set

$$\text{span} \left( \begin{bmatrix} \omega^{\sigma(1)} \\ \vdots \\ \omega^{\sigma(n)} \end{bmatrix} \,\middle|\, \sigma \in \mathcal{P}(n) \right) = \left\{ b \,\middle|\, \sum_{i=1}^n b_i = 0 \right\}.$$

In particular, it must hold that, for any $j$

$$X_j u - \frac{1}{n} \sum_{i=1}^n X_i u = 0.$$

Now adding our earlier derivation that

$$\left( \frac{1}{n} \sum_{i=1}^n X_i \right) u = \lambda u$$

gives us

$$X_j u = \lambda u,$$

which proves the lemma. $\qquad\square$

Next, we prove Lemma 3, the binomial theorem for random reshuffling.

**Lemma 3 (Binomial Theorem for Random Reshuffling)** For any symmetric matrices $X_1, \ldots, X_n$, and any constants $\alpha$ and $\beta$,

$$\text{rr}_k(\alpha I + \beta X_1, \alpha I + \beta X_2, \ldots, \alpha I + \beta X_n) = \sum_{i=0}^k \binom{k}{i} \alpha^{k-i} \beta^k \text{rr}_i(X_1, X_2, \ldots, X_n).$$

*Proof.* First, note that for any $A_1, A_2, \ldots, A_n$, we can write

$$\text{rr}_k(A_1, A_2, \ldots, A_n) = \frac{1}{n!} \frac{\partial}{\partial \eta_1} \frac{\partial}{\partial \eta_2} \cdots \frac{\partial}{\partial \eta_n} \left( \sum_{i=1}^n \eta_i A_i \right)^k \cdot \left( \sum_{i=1}^n \eta_i \right)^{n-k}.$$

This holds because

$$\left( \sum_{i=1}^n \eta_i A_i \right)^k \cdot \left( \sum_{i=1}^n \eta_i \right)^{n-k} = \sum_{\sigma:\{1,\ldots,n\}\to\{1,\ldots,n\}} \left( \prod_{i=1}^k \eta_{\sigma(i)} A_{\sigma(i)} \right) \left( \prod_{i=k+1}^n \eta_i \right),$$

where this sum ranges over all functions $\sigma$ from $\{1, \ldots, n\}$ to itself. The only terms of this sum that will survive the differentiation operation are those for which $\sigma$ is a permutation (since otherwise there will be some $\eta_i$ that does not appear in the term, and it will be zeroed out by the partial differentiation with respect to that $\eta_i$). So,

$$\frac{1}{n!} \frac{\partial}{\partial \eta_1} \frac{\partial}{\partial \eta_2} \cdots \frac{\partial}{\partial \eta_n} \left( \sum_{i=1}^{n} \eta_i A_i \right)^k \cdot \left( \sum_{i=1}^{n} \eta_i \right)^{n-k} = \frac{1}{n!} \sum_{\sigma \in \mathcal{P}(n)} \prod_{i=1}^{k} (\alpha I + \beta X_i)$$

which is $\mathrm{rr}_k(A_1, A_2, \ldots, A_n)$ by definition.

Applying this to our expression in the lemma statement,

$$\mathrm{rr}_k(\alpha I + \beta X_1, \ldots, \alpha I + \beta X_n) = \frac{1}{n!} \frac{\partial}{\partial \eta_1} \cdots \frac{\partial}{\partial \eta_n} \left( \sum_{i=1}^{n} \eta_i (\alpha I + \beta X_i) \right)^k \cdot \left( \sum_{i=1}^{n} \eta_i \right)^{n-k}$$

$$= \frac{1}{n!} \frac{\partial}{\partial \eta_1} \cdots \frac{\partial}{\partial \eta_n} \left( \alpha \sum_{i=1}^{n} \eta_i I + \beta \sum_{i=1}^{n} \eta_i X_i \right)^k \cdot \left( \sum_{i=1}^{n} \eta_i \right)^{n-k}$$

Applying the binomial theorem to this inner term gives us

$$\left( \alpha \sum_{i=1}^{n} \eta_i I + \beta \sum_{i=1}^{n} \eta_i X_i \right)^k = \sum_{j=0}^{k} \binom{k}{j} \left( \alpha \sum_{i=1}^{n} \eta_i I \right)^{k-j} \left( \beta \sum_{i=1}^{n} \eta_i X_i \right)^{j}$$

$$= \sum_{j=0}^{k} \binom{k}{j} \alpha^{k-j} \beta^k \left( \sum_{i=1}^{n} \eta_i \right)^{k-j} \left( \sum_{i=1}^{n} \eta_i X_i \right)^{j}.$$

Substituting this into our expression above gives

$$\mathrm{rr}_k(\alpha I + \beta X_1, \ldots, \alpha I + \beta X_n) = \frac{1}{n!} \frac{\partial}{\partial \eta_1} \cdots \frac{\partial}{\partial \eta_n} \sum_{j=0}^{k} \binom{k}{j} \alpha^{k-j} \beta^k \left( \sum_{i=1}^{n} \eta_i X_i \right)^{j} \cdot \left( \sum_{i=1}^{n} \eta_i \right)^{n-j}$$

$$= \sum_{j=0}^{k} \binom{k}{j} \alpha^{k-j} \beta^k \frac{1}{n!} \frac{\partial}{\partial \eta_1} \cdots \frac{\partial}{\partial \eta_n} \left( \sum_{i=1}^{n} \eta_i X_i \right)^{j} \cdot \left( \sum_{i=1}^{n} \eta_i \right)^{n-j}$$

$$= \sum_{j=0}^{k} \binom{k}{j} \alpha^{k-j} \beta^k \mathrm{rr}_j(X_1, \ldots, X_n).$$

This proves the lemma. $\qquad \square$

### F.2 Main Body of Proof of Theorem 1

In this section, we prove Theorem 1, which shows that Conjecture 1 holds with strict inequality for sufficiently small step sizes (as long as the matrix ensemble is nontrivial).

**Theorem 1 (Matrix AMGM Inequality, Sufficiently Small Step Size)** *Let $A_1, \ldots, A_n$ be a collection of continuously twice-differentiable functions from $\mathbb{R}$ to $\mathcal{S}_d$, that all satisfy $A_i(0) = I$ and that are non-trivial in the sense that they have no eigenvalue/eigenvector pairs shared among all the matrices $A_1'(0), A_2'(0), \ldots, A_n'(0)$. Then for any $2 \le k \le n$, there exists an $\alpha_{\max} > 0$ and a constant $C > 0$ such that such that for all $0 < \alpha \le \alpha_{\max}$,*

$$\|\mathrm{rr}_k(A_1(\alpha), A_2(\alpha), \ldots, A_n(\alpha))\| < \|\mathrm{rr}_1(A_1(\alpha), A_2(\alpha), \ldots, A_n(\alpha))\|^k - \alpha^2 C.$$

*Proof.* First, note that since when $\alpha = 0$,

$$\mathrm{rr}_k(A_1(\alpha), A_2(\alpha), \ldots, A_n(\alpha)) = \mathrm{rr}_1(A_1(\alpha), A_2(\alpha), \ldots, A_n(\alpha)) = I,$$

by continuity of these expressions in $\alpha$, for sufficiently small positive $\alpha$ it must hold that both

$$\mathsf{rr}_k(A_1(\alpha), A_2(\alpha), \ldots, A_n(\alpha))$$

and

$$\mathsf{rr}_1(A_1(\alpha), A_2(\alpha), \ldots, A_n(\alpha))$$

are positive definite. We will restrict our attention to $\alpha$ in this range (by setting $\alpha_{\max}$ appropriately). Next, define $\beta > 0$ as some number such that for any $\alpha$ in this range,

$$\frac{A_i(\alpha) - I}{\alpha} + \beta I \succeq 0.$$

The existence of such a $\beta$ is guaranteed because $A_i$ is continuously differentiable. Define

$$H_i(\alpha) = \frac{A_i(\alpha) - I}{\alpha} + \beta I.$$

Notice now that

$$A_i(\alpha) = \alpha H_i + (1 - \alpha\beta)I$$

So, by Lemma 3,

$$\mathsf{rr}_k(A_1(\alpha), A_2(\alpha), \ldots, A_n(\alpha)) = \mathsf{rr}_k(\alpha H_1(\alpha) + (1 - \alpha\beta)I, \ldots, \alpha H_n(\alpha) + (1 - \alpha\beta)I)$$

$$= \sum_{i=0}^{k} \binom{k}{i} \alpha^i (1 - \alpha\beta)^{k-i} \mathsf{rr}_i(H_1(\alpha), \ldots, H_n(\alpha)).$$

On the other hand, on the right side, by the binomial theorem,

$$(\mathsf{rr}_1(A_1(\alpha), A_2(\alpha), \ldots, A_n(\alpha)))^k = (\mathsf{rr}_1(\alpha H_1 + (1 - \alpha\beta)I, \ldots, \alpha H_n + (1 - \alpha\beta)I))^k$$

$$= \left( (1 - \alpha\beta)I + \frac{\alpha}{n} \sum_{i=1}^{n} H_i(\alpha) \right)^k$$

$$= \sum_{i=0}^{k} \binom{k}{i} \alpha^i \cdot (1 - \alpha\beta)^{k-i} \cdot \left( \frac{1}{n} \sum_{i=1}^{n} H_i(\alpha) \right)^k$$

$$= \sum_{i=0}^{k} \binom{k}{i} \alpha^i \cdot (1 - \alpha\beta)^{k-i} \cdot (\mathsf{rr}_1(H_1(\alpha), \ldots, H_n(\alpha)))^k.$$

Next, note that we can complete $H_i$ to be a continuously differentiable function by defining

$$H_i(0) = \lim_{\alpha \to 0} H_i(\alpha) = A_i'(0) + \beta I.$$

This means that, by our assumption, $H_1(0), H_2(0), \ldots, H_n(0)$ do not share any eigenvalue/eigenvector pairs. Therefore, by Lemma 4, we know that for any $u$

$$u^T \mathsf{rr}_2(H_1(0), H_2(0), \ldots, H_n(0))u < \|\mathsf{rr}_1(H_1(0), H_2(0), \ldots, H_n(0))\|^2,$$

(since this can not hold with equality because of our assumption about the eigenvectors of the $H_i$). Define $b$ as the constant such that

$$2b = \|\mathsf{rr}_1(H_1(0), H_2(0), \ldots, H_n(0))\|^2 - \lambda_{\max}(\mathsf{rr}_2(H_1(0), H_2(0), \ldots, H_n(0))).$$

We know that $b > 0$ because of our result from Lemma 4. By continuity of the $H_i$ and all these other expressions, for all sufficiently small $\alpha$,

$$\|\mathsf{rr}_1(H_1(\alpha), H_2(\alpha), \ldots, H_n(\alpha))\|^2 - \lambda_{\max}(\mathsf{rr}_2(H_1(\alpha), H_2(\alpha), \ldots, H_n(\alpha))) \geq b.$$

We will restrict our attention to $\alpha$ such that this holds (again by setting $\alpha_{\max}$ appropriately). In a similar way, we also restrict our attention to $\alpha$ such that $1 - \alpha\beta > 0$.

Let $c$ be a constant such that for any $i \in \{1, \ldots, n\}$, and any $\alpha$ in our range.

$$\|\mathsf{rr}_i(H_1(\alpha), \ldots, H_n(\alpha))\| \leq c^i$$

Such a $c$ must exist because it is a bound on a continuous function over a bounded interval. For any unit vector $u$, and any $\alpha > 0$ in our restricted sufficiently-small range

$$u^T \mathsf{rr}_k(A_1(\alpha), \ldots, A_n(\alpha)) u$$

$$= \sum_{i=0}^{k} \binom{k}{i} \alpha^i (1 - \alpha\beta)^{k-i} u^T \mathsf{rr}_i(H_1(\alpha), \ldots, H_n(\alpha)) u$$

$$= (1 - \alpha\beta)^k + \binom{n}{1} \cdot \alpha \cdot (1 - \alpha\beta)^{k-1} \cdot u^T \mathsf{rr}_1(H_1(\alpha), \ldots, H_n(\alpha)) u$$

$$+ \binom{n}{2} \cdot \alpha^2 \cdot (1 - \alpha\beta)^{k-2} \cdot u^T \mathsf{rr}_2(H_1(\alpha), \ldots, H_n(\alpha)) u$$

$$+ \sum_{i=3}^{k} \binom{k}{i} \cdot \alpha^i (1 - \alpha\beta)^{k-i} u^T \mathsf{rr}_i(H_1(\alpha), \ldots, H_n(\alpha)) u$$

$$\leq (1 - \alpha\beta)^k + \binom{n}{1} \cdot \alpha \cdot (1 - \alpha\beta)^{k-1} \cdot \|\mathsf{rr}_1(H_1(\alpha), \ldots, H_n(\alpha))\|$$

$$+ \binom{n}{2} \cdot \alpha^2 \cdot (1 - \alpha\beta)^{k-2} \cdot \left( \|\mathsf{rr}_1(H_1(\alpha), \ldots, H_n(\alpha))\|^2 - b \right)$$

$$+ \sum_{i=3}^{k} \binom{k}{i} \cdot \alpha^i (1 - \alpha\beta)^{k-i} \|\mathsf{rr}_i(H_1(\alpha), \ldots, H_n(\alpha))\|$$

$$\leq (1 - \alpha\beta)^k + \binom{n}{1} \cdot \alpha \cdot (1 - \alpha\beta)^{k-1} \cdot \|\mathsf{rr}_1(H_1(\alpha), \ldots, H_n(\alpha))\|$$

$$+ \binom{n}{2} \cdot \alpha^2 \cdot (1 - \alpha\beta)^{k-2} \cdot \left( \|\mathsf{rr}_1(H_1(\alpha), \ldots, H_n(\alpha))\|^2 - b \right)$$

$$+ \sum_{i=3}^{k} \binom{k}{i} \cdot \alpha^i \cdot (1 - \alpha\beta)^{k-i} \cdot c^i$$

$$\leq (1 - \alpha\beta)^k + \binom{n}{1} \cdot \alpha \cdot (1 - \alpha\beta)^{k-1} \cdot \|\mathsf{rr}_1(H_1(\alpha), \ldots, H_n(\alpha))\|$$

$$+ \binom{n}{2} \cdot \alpha^2 \cdot (1 - \alpha\beta)^{k-2} \cdot \left( \|\mathsf{rr}_1(H_1(\alpha), \ldots, H_n(\alpha))\|^2 - b \right)$$

$$+ \sum_{i=3}^{k} \binom{k}{i} \cdot \alpha^i \cdot (1 - \alpha\beta)^{k-i} \cdot \left( c^i + \|\mathsf{rr}_1(H_1(\alpha), \ldots, H_n(\alpha))\|^i \right)$$

$$\leq \left( (1 - \alpha\beta) + \alpha \|\mathsf{rr}_1(H_1(\alpha), \ldots, H_n(\alpha))\| \right)^k$$

$$- \binom{n}{2} \cdot \alpha^2 \cdot (1 - \alpha\beta)^{k-2} \cdot b$$

$$+ \sum_{i=3}^{k} \binom{k}{i} \cdot \alpha^i \cdot (1 - \alpha\beta)^{k-i} \cdot c^i.$$

Since all the $H_i$ are positive semidefinite,

$$(1 - \alpha\beta) + \alpha \|\mathsf{rr}_1(H_1(\alpha), \ldots, H_n(\alpha))\| = (1 - \alpha\beta) + \frac{\alpha}{n} \left\| \sum_{i=1}^{n} H_i(\alpha) \right\|$$

$$= \left\| (1 - \alpha\beta)I + \frac{\alpha}{n} \sum_{i=1}^{n} H_i(\alpha) \right\|$$

$$= \|\mathsf{rr}_1(A_1(\alpha), \ldots, A_n(\alpha))\| .$$

So, for any unit vector $u$,

$$u^T \mathrm{rr}_k(A_1(\alpha), \ldots, A_n(\alpha))u \le \|\mathrm{rr}_1(A_1(\alpha), \ldots, A_n(\alpha))\|^k$$
$$- \binom{n}{2} \cdot \alpha^2 \cdot (1 - \alpha\beta)^{k-2} \cdot b$$
$$+ \sum_{i=3}^{k} \binom{k}{i} \cdot \alpha^i \cdot (1 - \alpha\beta)^{k-i} \cdot c^i.$$

Since we chose $\alpha$ small enough that $\mathrm{rr}_k(A_1(\alpha), \ldots, A_n(\alpha))$ is positive semidefinite, and for any positive semidefinite matrix $X$ there exists a unit vector $u$ such that $\|X\| = u^T X u$, it follows that

$$\|\mathrm{rr}_k(A_1(\alpha), \ldots, A_n(\alpha))\| \le \|\mathrm{rr}_1(A_1(\alpha), \ldots, A_n(\alpha))\|^k$$
$$- \binom{n}{2} \cdot \alpha^2 \cdot (1 - \alpha\beta)^{k-2} \cdot b$$
$$+ \sum_{i=3}^{k} \binom{k}{i} \cdot \alpha^i \cdot (1 - \alpha\beta)^{k-i} \cdot c^i.$$

Since the expression

$$-\binom{n}{2} \cdot \alpha^2 \cdot (1 - \alpha\beta)^{k-2} \cdot b + \sum_{i=3}^{k} \binom{k}{i} \cdot \alpha^i \cdot (1 - \alpha\beta)^{k-i} \cdot c^i = -\frac{n(n-1)b}{2} \cdot \alpha^2 + \mathcal{O}(\alpha^3),$$

it follows that for sufficiently small $\alpha$,

$$-\binom{n}{2} \cdot \alpha^2 \cdot (1 - \alpha\beta)^{k-2} \cdot b + \sum_{i=3}^{k} \binom{k}{i} \cdot \alpha^i \cdot (1 - \alpha\beta)^{k-i} \cdot c^i < -\frac{n(n-1)b}{4} \cdot \alpha^2.$$

Now letting $C = \frac{n(n-1)b}{4}$ and letting $\alpha_{\max}$ denote the smallest bound on $\alpha$ we have assumed during this proof (including the "sufficiently small" bound we just invoked), it follows that

$$\|\mathrm{rr}_k(A_1(\alpha), \ldots, A_n(\alpha))\| < \|\mathrm{rr}_1(A_1(\alpha), \ldots, A_n(\alpha))\|^k - \alpha^2 C.$$

This is what we wanted to prove. $\qquad\square$

## G  Proof of Corollary 1

In this section, we prove Corollary 1, which characterizes the convergence of with-replacement versus without-replacement parallel SGD when a diminishing step size scheme is used.

**Corollary 1**  *Consider Algorithm 1 on a non-trivial noiseless convex quadratic using any $M$ (including $M = 1$, ordinary sequential SGD). Suppose that the step size scheme satisfies $0 < \alpha_i L < 1$ and is diminishing but not square-summable, i.e. $\lim_{k\to\infty} \alpha_k = 0$ and $\sum_{k=1}^{\infty} \alpha_k^2 = \infty$. Then for almost all initial values $w_0 \ne w^*$,*

$$\lim_{k\to\infty} \frac{\|\mathbf{E}[w_{k,without\text{-}replacement}] - w^*\|}{\|\mathbf{E}[w_{k,with\text{-}replacement}] - w^*\|} = 0.$$

*Proof.* Without loss of generality, suppose $w^* = 0$. Also for simplicity let $m = 1$ (referring to the one parallel worker in the algorithm, since $M = 1$). In this case, our loss functions are

$$f_i(w) = \frac{1}{2} w^T H_i w.$$

With this, the updates of Algorithm 1 can be written as

$$u_{k,m,t} = u_{k,m,t-1} - \alpha_k \nabla f_{\sigma_{k,m}(t)}(u_{k,m,t-1})$$
$$= u_{k,m,t-1} - \alpha_k H_{\sigma_{k,m}(t)} u_{k,m,t-1}$$
$$= \left(I - \alpha_k H_{\sigma_{k,m}(t)}\right) u_{k,m,t-1}.$$

Now, by induction,
$$u_{k,m,n} = \left(\prod_{t=1}^{n} \left(I - \alpha_k H_{\sigma_{k,m}(t)}\right)\right) w_k,$$
and so
$$w_k = \frac{1}{M} \sum_{m=1}^{M} \left(\prod_{t=1}^{n} \left(I - \alpha_k H_{\sigma_{k,m}(t)}\right)\right) w_{k-1}.$$
Now taking the expected value, for random reshuffling
$$\mathbf{E}\left[w_{k,\texttt{without-replacement}}\right] = \mathsf{rr}_n(I - \alpha_k H_1, \ldots, I - \alpha_k H_n) \cdot \mathbf{E}\left[w_{k-1,\texttt{without-replacement}}\right]$$
and for with-replacement sampling
$$\mathbf{E}\left[w_{k,\texttt{with-replacement}}\right] = \mathsf{rr}_1(I - \alpha_k H_1, \ldots, I - \alpha_k H_n)^n \cdot \mathbf{E}\left[w_{k-1,\texttt{with-replacement}}\right].$$
Let $\lambda$ be the smallest eigenvalue of
$$\mathsf{rr}_1(H_1, \ldots, H_n) = \frac{1}{n} \sum_{i=1}^{n} H_i$$
with corresponding eigenvalue $u$. It follows that
$$\mathsf{rr}_1(I - \alpha_k H_1, \ldots, I - \alpha_k H_n)u = (1 - \alpha_k \lambda)u = \|\mathsf{rr}_1(I - \alpha_k H_1, \ldots, I - \alpha_k H_n)\| u.$$
And so, for with-replacement sampling,
$$\left|u^T \mathbf{E}\left[w_{k,\texttt{with-replacement}}\right]\right| = \|\mathsf{rr}_1(I - \alpha_k H_1, \ldots, I - \alpha_k H_n)\|^n \cdot \left|u^T \mathbf{E}\left[w_{k-1,\texttt{with-replacement}}\right]\right|.$$
Applying this recursively,
$$\left|u^T \mathbf{E}\left[w_{K,\texttt{with-replacement}}\right]\right|$$
$$= \left|u^T \mathbf{E}\left[w_{0,\texttt{with-replacement}}\right]\right| \cdot \prod_{k=1}^{K} \|\mathsf{rr}_1(I - \alpha_k H_1, \ldots, I - \alpha_k H_n)\|^n.$$
For almost all initial values, $\left|u^T \mathbf{E}\left[w_{0,\texttt{with-replacement}}\right]\right| \neq 0$. Call this nonzero quantity $\rho$. Then by Cauchy-Schwarz,
$$\left\|\mathbf{E}\left[w_{K,\texttt{with-replacement}}\right]\right\| \geq \rho \cdot \prod_{k=1}^{K} \|\mathsf{rr}_1(I - \alpha_k H_1, \ldots, I - \alpha_k H_n)\|^n.$$
On the other hand, for random reshuffling
$$\left\|\mathbf{E}\left[w_{k+1,\texttt{without-replacement}}\right]\right\| \leq \|\mathsf{rr}_n(I - \alpha_k H_1, \ldots, I - \alpha_k H_n)\| \cdot \left\|\mathbf{E}\left[w_{k,\texttt{without-replacement}}\right]\right\|.$$
By Theorem 1, there exists some $\alpha_{\max} > 0$ and some $C$ such that for all $0 < \alpha < \alpha_{\max}$,
$$\|\mathsf{rr}_1(I - \alpha H_1, \ldots, I - \alpha H_n)\| < \|\mathsf{rr}_1(I - \alpha H_1, \ldots, I - \alpha H_n)\|^n - \alpha^2 C.$$
Since $\lim_{k \to \infty} \alpha_k = 0$, there exists some $\kappa$ such that for all $k \geq \kappa$, $\alpha_k < \alpha_{\max}$. So, we can bound our random-reshuffled distance recursively, for any $K > \kappa$ as
$$\left\|\mathbf{E}\left[w_{K,\texttt{without-replacement}}\right]\right\|$$
$$\leq \left\|\mathbf{E}\left[w_{0,\texttt{without-replacement}}\right]\right\| \cdot \prod_{k=1}^{K} \|\mathsf{rr}_n(I - \alpha_k H_1, \ldots, I - \alpha_k H_n)\|$$
$$= \left\|\mathbf{E}\left[w_{0,\texttt{without-replacement}}\right]\right\| \cdot \prod_{k=1}^{\kappa-1} \|\mathsf{rr}_n(I - \alpha_k H_1, \ldots, I - \alpha_k H_n)\|$$
$$\cdot \prod_{k=\kappa}^{K} \|\mathsf{rr}_n(I - \alpha_k H_1, \ldots, I - \alpha_k H_n)\|$$
$$\leq \left\|\mathbf{E}\left[w_{0,\texttt{without-replacement}}\right]\right\| \cdot \prod_{k=1}^{\kappa-1} \|\mathsf{rr}_n(I - \alpha_k H_1, \ldots, I - \alpha_k H_n)\|$$
$$\cdot \prod_{k=\kappa}^{K} \left(\|\mathsf{rr}_1(I - \alpha_k H_1, \ldots, I - \alpha_k H_n)\|^n - \alpha_k^2 C\right)$$

If we let $\left\|\mathbf{E}\left[w_{0,\texttt{without-replacement}}\right]\right\| = \phi$ and divide this by the with-replacement term, we get

$$\frac{\left\|\mathbf{E}\left[w_{K,\texttt{without-replacement}}\right]\right\|}{\left\|\mathbf{E}\left[w_{K,\texttt{with-replacement}}\right]\right\|}$$

$$\leq \frac{\phi}{\rho} \cdot \prod_{k=1}^{\kappa-1} \frac{\|\mathsf{rr}_n(I - \alpha_k H_1, \ldots, I - \alpha_k H_n)\|}{\|\mathsf{rr}_1(I - \alpha_k H_1, \ldots, I - \alpha_k H_n)\|^n}$$

$$\cdot \prod_{k=\kappa}^{K} \left(1 - \frac{\alpha_k^2 C}{\|\mathsf{rr}_1(I - \alpha_k H_1, \ldots, I - \alpha_k H_n)\|^n}\right)$$

Since each $H_i$ is positive semidefinite, and each $\alpha_k$ is sufficiently small, it follows that

$$\|\mathsf{rr}_1(I - \alpha_k H_1, \ldots, I - \alpha_k H_n)\| \leq 1.$$

So,

$$\frac{\left\|\mathbf{E}\left[w_{K,\texttt{without-replacement}}\right]\right\|}{\left\|\mathbf{E}\left[w_{K,\texttt{with-replacement}}\right]\right\|}$$

$$\leq \frac{\phi}{\rho} \cdot \prod_{k=1}^{\kappa-1} \frac{\|\mathsf{rr}_n(I - \alpha_k H_1, \ldots, I - \alpha_k H_n)\|}{\|\mathsf{rr}_1(I - \alpha_k H_1, \ldots, I - \alpha_k H_n)\|^n}$$

$$\cdot \prod_{k=\kappa}^{K} \left(1 - \alpha_k^2 C\right)$$

$$\leq \frac{\phi}{\rho} \cdot \prod_{k=1}^{\kappa-1} \frac{\|\mathsf{rr}_n(I - \alpha_k H_1, \ldots, I - \alpha_k H_n)\|}{\|\mathsf{rr}_1(I - \alpha_k H_1, \ldots, I - \alpha_k H_n)\|^n} \cdot \exp\left(-\sum_{k=\kappa}^{K} \alpha_k^2 C\right).$$

Since by assumption this sum diverges as $K \to \infty$, it follows that this entire expression converges to 0. This is what we wanted to show. $\qquad\square$

# H  Code to Reproduce Figures

Here, for completeness, and because the code is relatively short, we provide Julia code to reproduce the figures in this paper.

## H.1  Code to Reproduce Figure 1

```julia
using Random
using Statistics
using LinearAlgebra
using PyPlot

# Code to implement parallel SGD

abstract type SampleStrategy end;
abstract type WithReplacement <: SampleStrategy end;
abstract type WithoutReplacement <: SampleStrategy end;

function grad_fi(w::Array{Float64,1}, u::Array{Float64,1}, v::Array{Float64,1},
↪   a::Float64, gamma::Float64)
    return 2 * (dot(u,w)*dot(v,w) - a) * (dot(u,w)*v + dot(v,w)*u) + gamma * w;
end

function par_sgd_epoch(w0::Array{Float64,1}, us::Array{Array{Float64,1},1},
↪   vs::Array{Array{Float64,1},1}, as::Array{Float64,1}, alpha::Float64,
↪   gamma::Float64, ::Type{WithReplacement})
    w = w0;
    n = length(as);
```

```
19      for t = 1:n
20          i = rand(1:n);
21          w = w - alpha * grad_fi(w, us[i], vs[i], as[i], gamma);
22      end
23      return w;
24  end
25
26  function par_sgd_epoch(w0::Array{Float64,1}, us::Array{Array{Float64,1},1},
    ↪  vs::Array{Array{Float64,1},1}, as::Array{Float64,1}, alpha::Float64,
    ↪  gamma::Float64, ::Type{WithoutReplacement})
27      w = w0;
28      n = length(as);
29      s = randperm(n)
30      for t = 1:n
31          i = s[t];
32          w = w - alpha * grad_fi(w, us[i], vs[i], as[i], gamma);
33      end
34      return w;
35  end
36
37  function par_sgd(w0::Array{Float64,1}, us::Array{Array{Float64,1},1},
    ↪  vs::Array{Array{Float64,1},1}, as::Array{Float64,1}, alpha::Float64,
    ↪  gamma::Float64, M::Int64, K::Int64, ::Type{RS}) where {RS<:SampleStrategy}
38      w = w0;
39      ws = [copy(w0)];
40      for k = 1:K
41          w = mean([par_sgd_epoch(w, us, vs, as, alpha, gamma, RS) for m = 1:M]);
42          # println("at iter k, norm(w) = (norm(w))");
43          push!(ws, w);
44      end
45      return ws;
46  end
47
48  function mc_loss(w::Array{Float64,1}, us::Array{Array{Float64,1},1},
    ↪  vs::Array{Array{Float64,1},1}, as::Array{Float64,1}, gamma::Float64)
49      n = length(as);
50      return mean((dot(w,us[i])*dot(w,vs[i]) - as[i])^2 for i = 1:n) + gamma *
        ↪  norm(w)^2 / 2;
51  end
52
53
54  # Parameters
55
56  n = 40;
57  alpha = 0.1;
58  gamma = 0.05;
59  y_ii = sqrt(n-1)/n;
60  y_ij = -1.0/(n*sqrt(n-1));
61
62  us = [ones(n) for i = 1:n];
63  vs = [[(i == j) ? y_ii : y_ij for j = 1:n] for i = 1:n];
64  as = [(1 - alpha * gamma)/(2 * alpha) for i = 1:n];
65
66  M = 1000;
67  K = 100;
68
69  # Run experiments and produce figure
70
71  Random.seed!(1234567);
72
73  w0 = randn(n); w0 = w0 / norm(w0);
74
75  optimal_loss = mc_loss(zeros(n),us,vs,as,gamma);
76
77  figure(figsize=(4.0, 2.75));
```

```
78
79  for RUN = 1:10
80      w0 = 0.1 * randn(n);
81      ws_with = par_sgd(w0, us, vs, as, alpha, gamma, M, K, WithReplacement);
82      ws_without = par_sgd(w0, us, vs, as, alpha, gamma, M, K, WithoutReplacement);
83      losses_with = [mc_loss(w,us,vs,as,gamma) for w in ws_with];
84      losses_without = [mc_loss(w,us,vs,as,gamma) for w in ws_without];
85      if RUN == 10
86          semilogy(losses_with .- optimal_loss; c="#1f77a4", label="with
            ↪  replacement");
87          semilogy(losses_without .- optimal_loss; c="#c62028", label="without
            ↪  replacement");
88      else
89          semilogy(losses_with .- optimal_loss; c="#1f77a4", linestyle=":");
90          semilogy(losses_without .- optimal_loss; c="#c62028", linestyle=":");
91      end
92  end
93
94  legend();
95  ylabel("loss gap");
96  xlabel("epoch number");
97
98  tight_layout();
99
100 savefig("parsgdexample.pdf");
```

## H.2   Code to Reproduce Figure 2

```
1  using Random
2  using Statistics
3  using LinearAlgebra
4  using PyPlot
5
6
7  abstract type SampleStrategy end;
8  abstract type WithReplacement <: SampleStrategy end;
9  abstract type WithoutReplacement <: SampleStrategy end;
10
11 function grad_fi(w::Array{Float64,1}, H::Array{Float64})
12     return H*w;
13 end
14
15 function par_sgd_epoch(w0::Array{Float64,1}, Hs::Array{Array{Float64,2},1},
   ↪  alpha::Float64, ::Type{WithReplacement})
16     w = w0;
17     n = length(Hs);
18     for t = 1:n
19         i = rand(1:n);
20         w = w - alpha * grad_fi(w, Hs[i]);
21     end
22     return w;
23 end
24
25 function par_sgd_epoch(w0::Array{Float64,1}, Hs::Array{Array{Float64,2},1},
   ↪  alpha::Float64, ::Type{WithoutReplacement})
26     w = w0;
27     n = length(Hs);
28     s = randperm(n)
29     for t = 1:n
30         i = s[t];
31         w = w - alpha * grad_fi(w, Hs[i]);
32     end
```

```julia
33      return w;
34  end
35
36  function par_sgd(w0::Array{Float64,1},  Hs::Array{Array{Float64,2},1},
    ↪    alpha::Float64, M::Int64, K::Int64, ::Type{RS}) where {RS<:SampleStrategy}
37      w = w0;
38      ws = [copy(w0)];
39      for k = 1:K
40          w = mean([par_sgd_epoch(w, Hs, alpha, RS) for m = 1:M]);
41          # println("at iter k,norm(w) =(norm(w))");
42          push!(ws, w);
43      end
44      return ws;
45  end
46
47  function blockdiag(Xs::Array{T}...) where {T<:Number}
48      m = sum(size(X,1) for X in Xs);
49      n = sum(size(X,2) for X in Xs);
50      rv = zeros(T,m,n);
51      i = 0;
52      j = 0;
53      for X in Xs
54          rv[i.+(1:size(X,1)),j.+(1:size(X,2))] .= X;
55          i += size(X,1);
56          j += size(X,2);
57      end
58      return rv;
59  end
60
61  # construct the example
62
63  n = 8
64
65  A_kij = -2/(n*sqrt(n-1));
66  A_kii = 1 - 2/(n*sqrt(n-1));
67  A_kki = (n-2)/(n*sqrt(n-1));
68  A_kkk = 1 + (2*sqrt(n-1))/n;
69
70  As = [[(i == j) ? ((i == k) ? A_kkk : A_kii) : (((i==k)(j==k)) ? A_kki : A_kij) for
    ↪    i = 1:n, j = 1:n] for k = 1:n];
71
72  beta = 0.5;
73
74  Hs = [blockdiag(I - beta * A, [1 - beta], [1 + beta]) for A in As];
75
76
77  # problem settings
78  M = 100;
79  K = 20;
80  alphas = 0.0:0.001:1.7;
81
82  # initial value
83  Random.seed!(8675309)
84  w0 = vcat(randn(n), 1e-20*randn(2)); # note smaller magnitude in extra component
85
86  # run experiments
87  y_with = [norm(par_sgd(w0, Hs, alpha, M, K, WithReplacement)[end]) for alpha in
    ↪    alphas];
88  y_without = [norm(par_sgd(w0, Hs, alpha, M, K, WithoutReplacement)[end]) for alpha
    ↪    in alphas];
89
90  y_with1 = [norm(par_sgd(w0, Hs, alpha, 1, K, WithReplacement)[end]) for alpha in
    ↪    alphas];
91  y_without1 = [norm(par_sgd(w0, Hs, alpha, 1, K, WithoutReplacement)[end]) for alpha
    ↪    in alphas];
```

```
92
93
94 figure(figsize=(4.0, 2.75));
95
96 semilogy(alphas, y_with1; label="with replacement (M=1)", c="#7fc7e4",
   ↪  linestyle=":");
97 semilogy(alphas, y_without1; label="without replacement (M=1)", c="#e6a798",
   ↪  linestyle=":");
98 semilogy(alphas, y_with; label="with replacement (M=100)", c="#1f77a4");
99 semilogy(alphas, y_without; label="without replacement (M=100)", c="#c62028");
100 xlim((0.0,1.7));
101 xlabel("step size");
102 ylabel("distance to optimum");
103 legend(frameon=false, fontsize="small");
104
105 tight_layout();
106
107 savefig("convexsgdexample.pdf");
```

## H.3 Code to Reproduce Figure 3

```
1 using LinearAlgebra
2 using Random
3 using PyPlot
4
5 A1 = BigInt.([2 -1 1; -1 2 -1; 1 -1 1]).//4
6 A2 = BigInt.([2 0 -1; 0 1 1; -1 1 2]).//4
7 A3 = BigInt.([0 0 0; 0 1 -1; 0 -1 2]).//4
8 u = BigInt.([1, -2, -1]).//1;
9 R = (u*u')//6;
10
11 # assert that the product of the matrices is in fact nilpotent
12 @assert((A1*A2*A3)^3 == zeros(Rational{BigInt},3,3))
13
14 # check that all the matrices are in fact positive semidefinite
15 @assert(all(eigvals(Float64.(A1)) .>= 0))
16 @assert(all(eigvals(Float64.(A2)) .>= 0))
17 @assert(all(eigvals(Float64.(A3)) .>= 0))
18 @assert(all(eigvals(Float64.(R)) .>= -1e-8))
19
20 # check that all the matrices are in fact all \preceq I
21 @assert(all(eigvals(Float64.(A1)) .<= 1))
22 @assert(all(eigvals(Float64.(A2)) .<= 1))
23 @assert(all(eigvals(Float64.(A3)) .<= 1))
24 @assert(all(eigvals(Float64.(R)) .<= 1))
25
26 # all sequences of matrices that can occur in a single permutation of A1, A2, and
   ↪  A3
27 Ls = [BigInt.(Matrix(I,3,3)).//1,
28       A1,A2,A3,
29       A1*A2,A1*A3,A2*A1,A2*A3,A3*A1,A3*A2,
30       A1*A2*A3,A1*A3*A2,A2*A1*A3,A2*A3*A1,A3*A1*A2,A3*A2*A1];
31
32 # assert that no possible sequence can be 0
33 for L1 in Ls
34     for L2 in Ls
35         @assert(u'*L1*L2*u != 0)
36     end
37 end
38
39 # function for SGD using without-replacement sampling
```

```julia
40  function sgd_rr(x0::Array{Rational{BigInt},1},
     ↪  DS::Array{Array{Rational{BigInt},2},1}, alpha::Rational{BigInt},
     ↪  num_epochs::Int64)
41      rv = Float64[];
42      x = x0;
43      for iepoch = 1:num_epochs
44          for s in shuffle(DS)
45              x = x - alpha * 2 * s * x;
46          end
47          loss = sum(x'*s*x for s in DS)//length(DS);
48          push!(rv, Float64(log10(BigFloat(loss))));
49      end
50      return rv;
51  end
52
53  # function for SGD using with-replacement sampling
54  function sgd_wr(x0::Array{Rational{BigInt},1},
     ↪  DS::Array{Array{Rational{BigInt},2},1}, alpha::Rational{BigInt},
     ↪  num_epochs::Int64)
55      rv = Float64[];
56      x = x0;
57      for iepoch = 1:num_epochs
58          for i = 1:length(DS)
59              x = x - alpha * 2 * rand(DS) * x;
60          end
61          loss = sum(x'*s*x for s in DS)//length(DS);
62          push!(rv, Float64(log10(BigFloat(loss))));
63      end
64      return rv;
65  end
66
67  # random initial value
68  Random.seed!(8765309)
69  x0 = big.(Rational.(randn(Float32,3)))
70
71  # dataset
72  DS = [I - A1, I - A2, I - A3, I - R]
73
74  # step size
75  alpha = BigInt(1)//2
76
77  # number of epochs and trials to run
78  nepochs = 1000
79  ntrials = 20
80
81  # run experiments
82  log10_losses_wr = [sgd_wr(x0, DS, alpha, nepochs) for i = 1:ntrials]
83  log10_losses_rr = [sgd_rr(x0, DS, alpha, nepochs) for i = 1:ntrials]
84
85  # modify wo replacement experiments to show very small number instead of minus
     ↪  infinity
86  # this is an adjustment for display, since that small number will be out of the
     ↪  range of the axis
87  # we will get nice vertical lines in the plot slowing the with-replacement curve has
     ↪  jumped to 0
88  log10_losses_wr_adjusted = [[(x == -Inf) ? -1e8 : x for x in ld] for ld in
     ↪  log10_losses_wr]
89
90  # actually make the plot
91  figure(figsize=(4.0, 2.75));
92  plot(log10_losses_wr_adjusted[1]; label="with replacement", color="#1f77a4")
93  plot(log10_losses_rr[1]; label="without replacement", color="#c62028")
94  for dd in log10_losses_wr_adjusted[2:end]
95      plot(dd; color="#1f77a4", linestyle=":")
96  end
```

```
 97  for dd in log10_losses_rr[2:end]
 98      plot(dd; color="#c62028", linestyle=":")
 99  end
100  legend()
101  ylim([-4000,100])
102  xlabel("epoch number")
103  ylabel("\$\\log_{10}\$(loss)")
104  tight_layout()
105
106  savefig("asymptotic.pdf")
```