[Reviews · NeurIPS 2020]

Review 1

Summary and Contributions: Provides a counter-example for a key conjecture in optimization theory, and uses this example to construct an optimization problem where with-out replacement performs worse than with-replacement.

Strengths: This paper provides clear counter-examples to two variants to the Noncommutative Arithmetic and Geometric Mean inequality proposed by Recht and Ré. I find these examples quite clear and the paper generally easy to follow. Research on this topic is foundational for optimization, and I believe a better understanding of the phenomena of random-reshuffling output performing with-replacement is important and impactful. I don't full understand why it's necessary to use a parallel SGD example to illustrate the construction, I see that it averaging a large number of parallel steps should give a value closer to the expected value, is the variance otherwise two high with non-parallel SGD? Could that be fixed with larger n?

Weaknesses: A stronger counter-example for the plain SGD case would strengthen the paper further.

Correctness: Looks correct

Clarity: This paper is exceptionally clear and well written.

Relation to Prior Work: This paper comprehensively covers prior work.

Reproducibility: Yes

Additional Feedback: UPDATE: After discussion, based on the limitations of the examples constructions, I have revised my score down a little.


Review 2

Summary and Contributions: This work considers non-commutative matrix AM-GM inequality and its application towards analysis of SGD with Random Reshuffling vs with replacement sampling for quadratic optimization problems. Recent works have derived sharp rates for convergences of Random Reshuffling and with replacement sampling methods to show that RR outperforms with-replacement SGD in terms of worst-case bounds for general classes of functions. It was however, an open problem to show whether or not RR outperforms with-replacement SGD for every problem (as observed often in practice). This can be shown for quadratic optimization problems assuming non-commutative AM-GM conjecture. The main contribution of the current work is to obtain a simple counter example for this conjecture and hence construct a quadratic optimization problem where RR performs strictly worse that with-replacement SGD, hence solving an important open problem.

Strengths: 1. The results are simple and succinct. The claims appear to be correct based upon my reading of the main text. 2. Despite its simplicity, the contribution is novel and significant and answers an important open question which is practically relevant. 3. Since without-replacement sampling methods are ubiquitous in first order stochastic optimization, this is very relevant to NeurIPS ======== After discussions with other reviewers, I have decided to reduce the score from 7 to 6.

Weaknesses: It has been established that via. worst case bounds that RR outperforms with-replacement SGD for a large class of functions. The paper just constructs one instance where RR underperforms SGD. Even though it solves the problem at hand, it gives little or no intuition about the structure which causes this behavior. It would be better if a class of such problems is constructed, which may hopefully explain structure responsible for this phenomenon. Another weakness is the requirement that the number of matrices n to be equal to the dimension d. In linear regression, n is typically much larger than d. Therefore, this counter example seems artificial.

Correctness: The claims appear to be correct based on my evaluation of the main text.

Clarity: Yes, the paper is well written. The main ideas are easy to follow.

Relation to Prior Work: The work discusses most of the relevant prior contributions in this area and explains why it is novel. However, the following citations seem to be missing: [1] SGD without Replacement: Sharper Rates for General Smooth Convex Functions by Dheeraj Nagaraj, Praneeth Netrapalli and Prateek Jain (ICML 2019) [2] Closing the convergence gap of SGD without replacement by by Shashank Rajput, Anant Gupta and Dimitris Papailiopoulos (arXiv preprint)

Reproducibility: Yes

Additional Feedback:


Review 3

Summary and Contributions: The paper compares random reshuffling to with-replacement sampling for stochastic learning algorithms on noiseless problem. The authors provide some counterexamples to show that the performance of the with-replacement scheme has some benefits over the without-replacement one.

Strengths: N/A. It is unclear about the contribution of this paper.

Weaknesses: - The paper is not well written and organized. Therefore, it is not clear about its contribution. - The conditions may be not practical. - Hard to provide any meaningful evidence to show the real benefits for the machine learning problems.

Correctness: N/A

Clarity: The paper is not well organized and written and hard to follow

Relation to Prior Work: Not really

Reproducibility: No

Additional Feedback: 1) It is true that random reshuffling is not always better than the regular SGD. Note that the convergence rate of O ( 1 / t^2) for random shuffling should also depend on “n” while the one of O ( 1 / t ) for regular one does not. I believe that no one has shown the rigorous theory to show that random reshuffling is always better than the regular scheme. They may be able to show some advantages of the random reshuffling over the regular scheme in experiments. 2) The problem in Definition 1 is not really practical. Could you please provide some specific ML problems satisfying all the conditions? In other words, the problem consists of strongly convex objective function and L-smooth and convex (quadratic) component function f_i. Moreover, the optimal solution w_* for the objective function is also an optimal solution for all f_i. For this problem, it is well-known that SGD with the regular scheme would have linear convergence rate to the optimal solution. It is not surprise if SGD with replacement could be better than the random shuffling in this situation. 3) The paper is not well organized and written and hard to follow. I would suggest the authors to add some examples of ML models specifically and introduce the Definition 1 and its conditions in the beginning. The contribution of this paper is not clear since it is not able to conclude anything between the random shuffling and the regular with replacement scheme. At the current state, I think the paper is not ready for the publication at NeurIPS. ================= POST REBUTTAL ====================== I have read the author's response. However, it is still unclear to me why Conjectures 1 + 2, Theorem 1 and counterexamples in Section 2 show that Random Reshuffling under-performs regular SGD. The only things I have seen are the example in Section 3 and Corollary 1. However, this belongs to the class of Definition 1, which makes "the problem consists of strongly convex objective function and L-smooth and convex (quadratic) component function f_i. Moreover, the optimal solution w_* for the objective function is also an optimal solution for all f_i". For this problem, it is well-known that SGD with the regular scheme would have linear convergence rate to the optimal solution. Therefore, it seems clearly that it has better than the sub-linear rate of Random Reshuffling. I think it is not good enough if this is the main contribution of the paper.


Review 4

Summary and Contributions: This paper gives explicit, simple, counterexamples to three conjectured non-commutative generalizations of the AM-GM inequality that arise in comparing stochastic gradient descent strategies using sampling with replacement vs random reshuffling. Moreover, the paper then constructs an explicit example of an optimization problems in which an version of SGD (related to the conjectures) with randomized reshuffling converges more slowly than the SGD-variant for sampling wtih replacement. Since SGD usually incorporates a step size parameter, the paper then studies the effect of step size on the parameter. First it is shown that a variation on Conjectures 1-3 (Theorem 1) that captures the scenario of using sufficiently small step size, is in fact true. It follows (Corollary 1) that without replacement sampling always outperforms with replacement sampling on convex quadratics when both use the same step-size scheme (that is decaying and not square-summable). Interestingly, again via an explicit counterexample, it is shown that without the constraint that the two schemes use the same step size, it is again possible for sampling with replacement to outperform sampling without replacement. The paper concludes with an example showing that for vanilla SGD (without averaging) it is possible for with-replacement sampling to converge strictly faster than random reshuffling. == added in discussion phase == Upon reading the author responses and the other reviewers, I now see more clearly that the extensions of the counterexample to the Recht-Re conjecture to give optimization problems where one method outperforms another are perhaps a little artificial (both the problems and the methods). I do not think this, in any way, takes away from the importance of the counterexample (which is still a nice, fundamental, contribution), but does weaken a little claims to relevance of other parts of the paper.

Strengths: This paper studies a fundamental problem of significant theoretical interest to the community. It gives simple and clear counterexamples, that are also very clearly motivated, and shows that some folklore ideas relating randomized reshuffling vs sampling with replacement in SGD may not always be valid. It goes beyond studying just the underlying mathematical conjectures to clearly contextualize and extend the couterexamples to deal with a number of cases of more practical interest. I expect that some of these results and couterexamples will provide new intuition to the community about what makes these different sampling strategies work (or not work), and this new intuition may lead to new, deeper understanding of these sampling methods in SGD.

Weaknesses: One could perhaps argue that the algorithms considered (except in Section 5) are somewhat artificial variations on SGD. However, at least the results of section 5 suggest that it may be possible to construct such examples for many variants of SGD, and in the process understand issues of scan order much better.

Correctness: The paper appears to be correct. Moreover, the computational experiments appear to be carried out quite carefully (e.g., using exact arithmetic where needed).

Clarity: The paper is very clearly written. It uses (indeed is based on) very concrete examples. It very clearly motivates the choice of examples, explaining where they come from. There is also very clear explanation and motivation for the choice of learning problems discussed (based on the potential over-simplifications of the previous problems, slowly building in complexity).

Relation to Prior Work: The paper clearly and succinctly explains the origin of the conjectures that are studied, the progress made on them (up until very recent work), and how this work differs from previous papers that disprove some of the stated conjectures (explicit examples, explicit learning problems, etc.)

Reproducibility: Yes

Additional Feedback:

[Author Response · NeurIPS 2020]

*We thank all four reviewers for their insightful comments and helpful feedback.* Most of the reviewers describe the paper as very well written and clear, and most identified its contribution as being important to the ML community, as understanding scan orders is a fundamental problem in this space. We now address the reviewers' concerns individually.

**R1. Why is it necessary to use a parallel SGD example?** Good point! The parallelism itself is not necessary; what is necessary is the *averaging*. The averaging is necessary (even for large $n$) because it models the expected value in the original Recht and Ré inequality: without it the convergence rate may be effected by higher-order moments (not just the expected value). We study the non-averaged case in Section 5. We use parallel SGD as an example because it is a "real" algorithm that uses averaging (one that has been previously proposed in [9,26]). We will clarify this in the text.

**R2. What intuition is given about the structure which causes the behavior? Could a class of problems be constructed?** Part of our goal in Section 2 was to construct a class of multiple counterexamples of arbitrarily high dimension, to give better intuition. For example, in Section 2.1, our construction produces a counterexample whenever

$$|\lambda| = \left| \left(1 + \tfrac{1}{n-1}\right)^{n/2} \cdot \cos\left(n \cdot \arcsin\left(\tfrac{1}{\sqrt{n}}\right)\right) \right| > 1, \qquad \text{for example, when } 5 \leq n \leq 16 \text{ or } 29 \leq n \leq 51 \text{ etc.}$$

This family of counterexamples is inherited by our other counterexamples in 2.2 and 2.3, which depend on it. As far as we can tell, the "structure" responsible for this phenomenon (which allows us to get at the counterexamples) is the permutation-matrix-symmetry that we discuss at the beginning of Section 2.

**R2. Another weakness is the requirement that the number of matrices $n$ to be equal to the dimension $d$. In linear regression, n is typically much larger than d.** This can be avoided in the counterexample by simply adding additional $I$ matrices to the ensemble, since (1) this does not change the arithmetic mean, which is already $I$, and (2) this does not change the random-reshuffled product, which is not affected by multiplying by $I$. This approach can generate counterexamples with $n$ arbitrarily large compared to $d$. We will add text to clarify this.

**R2. Citations.** We thank R2 for the additional very helpful citations, which we will add to the paper.

**R4.** We thank Reviewer 4 for a clear and insightful review. We share R4's hopes that "counterexamples will provide new intuition to the community about what makes these different sampling strategies work (or not work), and this new intuition may lead to new, deeper understanding of these sampling methods in SGD," and we view this as a great summary of the impact we hope our results will have on the community.

**R3. Writing.** R3 says that they find the paper poorly written and difficult to follow. This writing issue seems to permeate and color the rest of R3's review: for example, they could not identify any strengths at all in the work, and say the contribution is unclear. We apologize for the confusion. This is perplexing to us because we are unsure what makes the writing unclear, and all the other reviewers said that the paper was clear, saying things like "very clear" and even "exceptionally clear." Perhaps our writing was insufficiently clear that main point of the paper is to present *negative results*, concrete counterexamples to longstanding conjectures in the space for which no previous constructive disproofs were known. We will try to improve this as we revise our manuscript.

**R3. Note that the convergence rate of $O(1/t^2)$ for random shuffling should also depend on "n" while the one of $O(1/t)$ for regular one does not.** Recent work ("SGD without Replacement: Sharper Rates for General Smooth Convex Functions," which Reviewer 2 just made us aware of) gives a convergence rate of $O(\frac{1}{nK^2})$ for $K$ epochs of random-reshuffling on $n$ examples, while the best result with replacement is $O(\frac{1}{nK})$. So while $n$ is involved, in this setting RR does indeed have an asymptotically better class upper bound. The fact that this problem-class upper bound does *not* transfer to RR being asymptotically better than with-replacement SGD for individual problems is interesting, and this is what we are trying to get at in this paper.

**R3. The problem in Definition 1 is not really practical. Could you provide some specific ML problems satisfying all the conditions?** This class roughly corresponds to the setting of the Randomized Kaczmarz method originally studied by Recht and Ré [17]. Here, the task is to solve a system of linear equations $Ax - b = 0$ (which for simplicity we assume has a unique solution) by minimizing the objective $\frac{1}{2n} \sum_{i=1}^{n} (a_i^T x - b_i)^2$, where $a_i$ are the rows of $A \in \mathbb{R}^{n \times d}$ and $b_i$ the entries of $b \in \mathbb{R}^n$. Since the problem has a unqiue solution $x^*$, the gradient of each component of this loss is 0 at that point, and they are of course obviously convex quadratics. It is fairly easy to choose $a_i$ such that the problem would be non-trivial. (See also our statement in Footnote 2 about why we did not try to generalize this class further.)

Also note that Definition 1 is only used to prove Corollary 1, which is introduced to give some context/interpretation for Theorem 1, which itself is the main technical result of the section—Definition 1 and the problem class it defines are not central to the claims of the paper.

**R3. Reproducibility and Broader Impact.** The code to reproduce all the figures in the paper is given in the supplemental material, as are all the proofs. As such, it is not clear to us why R3 evaluates the paper as not reproducible. We are equally confused about the "no" on broader impact, and would appreciate some feedback to improve that section.

[Meta-Review · NeurIPS 2020]

This paper received 4 reviews. Three of them consider the contribution of the paper are of significant interest to the ML community, and another reviewer thinks this paper does not give meaningful conclusion concerning comparison between with and without replacement sampling in SGD. I agree with the majority of the reviewers about the significance of the results on providing concrete counter examples for the two conjectures about matrix Arithmetic and Geometric mean inequalities, thus recommend acceptance of the paper. On the other hand, I agree with Reviewer #3 that the paper does not have concrete conclusion comparing with and without replacement sampling is SGD, and the paper's claim that "it is generally believed that ... random reshuffling causes learning algorithms to converge faster" is not precise. It is a phenomenon that people often observe, but I doubt many people has the belief that random reshuffling is "always better". The authors' effort in constructing counter examples to disprove the "general belief" is appreciated, but I think there are many like Reviewer #3 (including myself) having the experience that neither would always be better in practical SGD methods and considering the counter examples on SGD in sections 3-5 artificial and maybe unnecessary. I view the two opinions not necessarily contradicting each other. The conjectures on matrix arithmetic and geometric inequalities are abstractions from practical SGD variants on special loss functions. Construction of concrete counter examples for these mathematical conjectures is of major theoretical interest, but extending such constructions to demonstrate consequences for SGD in practice may involve other factors and less convincing. I highly recommend the authors to clarify such distinctions in the revision.